# Site-specific machine learning predictive fertilization models for potato crops in Eastern Canada

Zonlehoua Coulibali[1], Athyna Nancy Cambouris[2], Serge-Étienne Parent[1]*

1 Department of Soils and Agrifood Engineering, Université Laval, Québec City, Quebec, Canada, 2 Quebec Research and Development Centre, Agriculture and Agri-Food Canada, Québec City, Quebec, Canada

* serge-etienne.parent.1@ulaval.ca

**Data Availability Statement:** All relevant data are within the manuscript and its Supporting Information files. There is no restriction on sharing of data and/or materials.

## Abstract

Statistical modeling is commonly used to relate the performance of potato (*Solanum tuberosum* L.) to fertilizer requirements. Prescribing optimal nutrient doses is challenging because of the involvement of many variables including weather, soils, land management, genotypes, and severity of pests and diseases. Where sufficient data are available, machine learning algorithms can be used to predict crop performance. The objective of this study was to determine an optimal model predicting nitrogen, phosphorus and potassium requirements for high tuber yield and quality (size and specific gravity) as impacted by weather, soils and land management variables. We exploited a data set of 273 field experiments conducted from 1979 to 2017 in Quebec (Canada). We developed, evaluated and compared predictions from a hierarchical Mitscherlich model, *k*-nearest neighbors, random forest, neural networks and Gaussian processes. Machine learning models returned $R^2$ values of 0.49–0.59 for tuber marketable yield prediction, which were higher than the Mitscherlich model $R^2$ (0.37). The models were more likely to predict medium-size tubers ($R^2$ = 0.60–0.69) and tuber specific gravity ($R^2$ = 0.58–0.67) than large-size tubers ($R^2$ = 0.55–0.64) and marketable yield. Response surfaces from the Mitscherlich model, neural networks and Gaussian processes returned smooth responses that agreed more with actual evidence than discontinuous curves derived from *k*-nearest neighbors and random forest models. When conditioned to obtain optimal dosages from dose-response surfaces given constant weather, soil and land management conditions, some disagreements occurred between models. Due to their built-in ability to develop recommendations within a probabilistic risk-assessment framework, Gaussian processes stood out as the most promising algorithm to support decisions that minimize economic or agronomic risks.

## 1. Introduction

Modeling provides a quantitative understanding of how crop systems operate [1]. Site-specific simulations of fertilizer requirements to obtain high local potato yield and quality rely on models' ability to detect subtle variations in factors affecting plant growth and

**Funding:** ZC is partly funded by the Natural Sciences and Engineering Council of Canada (CRDPJ 385199-09 and DG-2254 - https://www.nserc-crsng.gc.ca), the Quebec Ministry of Agriculture, Fisheries and Food (IA216581 - https://www.mapaq.gouv.qc.ca), Centre SEVE (https://centreseve.recherche.usherbrooke.ca/), Patate Dolbec Inc. (https://patatesdolbec.com/), Groupe Gosselin FG (http://gosseling2.com), Agriparmentier Inc., Ferme Daniel Bolduc Inc. (http://fermedanielbolduc.com/), Patate Laurentienne, Ferme Bergeron-Niquet, and Patates Lac-St-Jean (http://plsj.ca/). There was no additional external funding received for this study. The funders had no role in study design, data collection and analysis, decision to publish, or preparation of the manuscript.

**Competing interests:** The authors have declared that no competing interests exist. All the funders (Natural Sciences and Engineering Council of Canada, Quebec Ministry of Agriculture, Fisheries and Food, Centre SEVE, Patate Dolbec Inc., Groupe Gosselin FG, Agriparmentier Inc., Patate Laurentienne, Ferme Bergeron-Niquet, and Patates Lac-St-Jean) have declared that no competing interests exist. This does not alter our adherence to PLOS ONE policies on sharing data and materials.

environment and to learn from the past to make predictions [2]. Several crop models have been developed with different degrees of sophistication, scale, and representativeness [2]. Mechanistic models have been published for potato cropping systems [3, 4]. Semi-mechanistic growth models could be used to downscale tuber yield assessment from regional to field levels [5, 6]. Multilevel modeling can assist in selecting a set of relevant parameters that impact tuber yield and fertilizer requirements, but can hardly predict site-specific nutrient requirements [7].

Several variables can impact fertilization at optimum tuber yield: soil type and quality [8, 9], organic fertilizers [10, 11], preceding crops [12–16], weather conditions [17], irrigation [18], timing, location and chemical form of the fertilizer applied [19], pests and diseases [20] and genetic factors such as cultivar longevity and growth rate [21, 22]. Air temperature, photoperiod, day length, intercepted radiation, water abundance, precipitations, root development and crop management were reported to be the driving variables for potato growth and development [8, 9, 23–26]. While the nitrogen (N) requirement of potato crops compares with other high N-demanding crops, phosphorus (P) uptake depends largely on close contact between roots and soil particles that, in turn, depends on soil texture, buffering capacity and moisture content [27, 28]. Due to a shallow system of fine roots and small biomass [29], especially in compacted soils [8, 9], potato is sensitive to nutrient and water stresses [30].

The N, P and K (potassium) requirements are thought to be cultivar- and market-specific [31–33]. Specific gravity (SG) is of particular concern for North-American processors [34, 35]. Other characteristics, such as tuber size and grade are also valued [34]. No model has yet addressed K requirements accounting for interactions between genetics, environment and management [36].

Growers tend to over-fertilize because of the potential economic loss from under-fertilizing [37, 38]. While N can cause nitrate contamination [39–42] and P the eutrophication of surface waters [43–45], K has no known deleterious effect on the quality of natural and drinking water. Attempts have been made to synthesize the results of fertilizer experiments using meta-analysis to derive N optima for specific soil texture and pH groups [46] or multilevel modeling combining soil, climate indices and management variables [7]. Even where field trials could identify nutrient optima [47], such optima cannot be generalized to conditions different from those of particular experiments [48, 49].

Although experimental data grow continuously in size and quality, it is still beyond researchers' ability to integrate, analyze and make the best-informed decisions. Machine learning is an emerging technology that can aid in the discovery of rules and patterns in large sets of data [50]. The technology bypasses intermediate processes otherwise explicitly explained by a mechanistic modeling system and makes predictions directly based on input data [51]. Machine learning methods can combine fertilizer dosage, genetics, environmental and land management variables to predict tuber yield and quality. Classical models such as Mitscherlich are limited to plant-nutrient relationships [52].

We hypothesized that (1) genetics, environment and local land management practices are the main drivers of fertilizer requirements, (2) *k*-nearest neighbors, random forest, neural networks and Gaussian processes are more accurate in predicting marketable yield than classical Mitscherlich predictive models, and (3) the machine learning algorithms are equally able to predict economic optimal or agronomic optimal fertilizer doses. The objective of this study was to develop, evaluate and compare the performance of machine learning models in predicting N, P and K requirements for potato.

## 2. Methodology

### 2.1 Data set

The Quebec (Canada) potato data set is a collection of field fertilizer trials conducted from 1979 to 2015 between the US border (45[th] parallel) and the Northern limit of cultivation (49[th] parallel). We added 17 trials conducted in 2016 and 2017. Fig 1 shows the location of experimental sites.

The trials with maximum yield less than 28 Mg ha⁻¹ were discarded to avoid extreme cases of diseases, management failures or catastrophic weather events. The data set contains 4254–5913 observations from 208–273 field trials, depending on the number of missing values

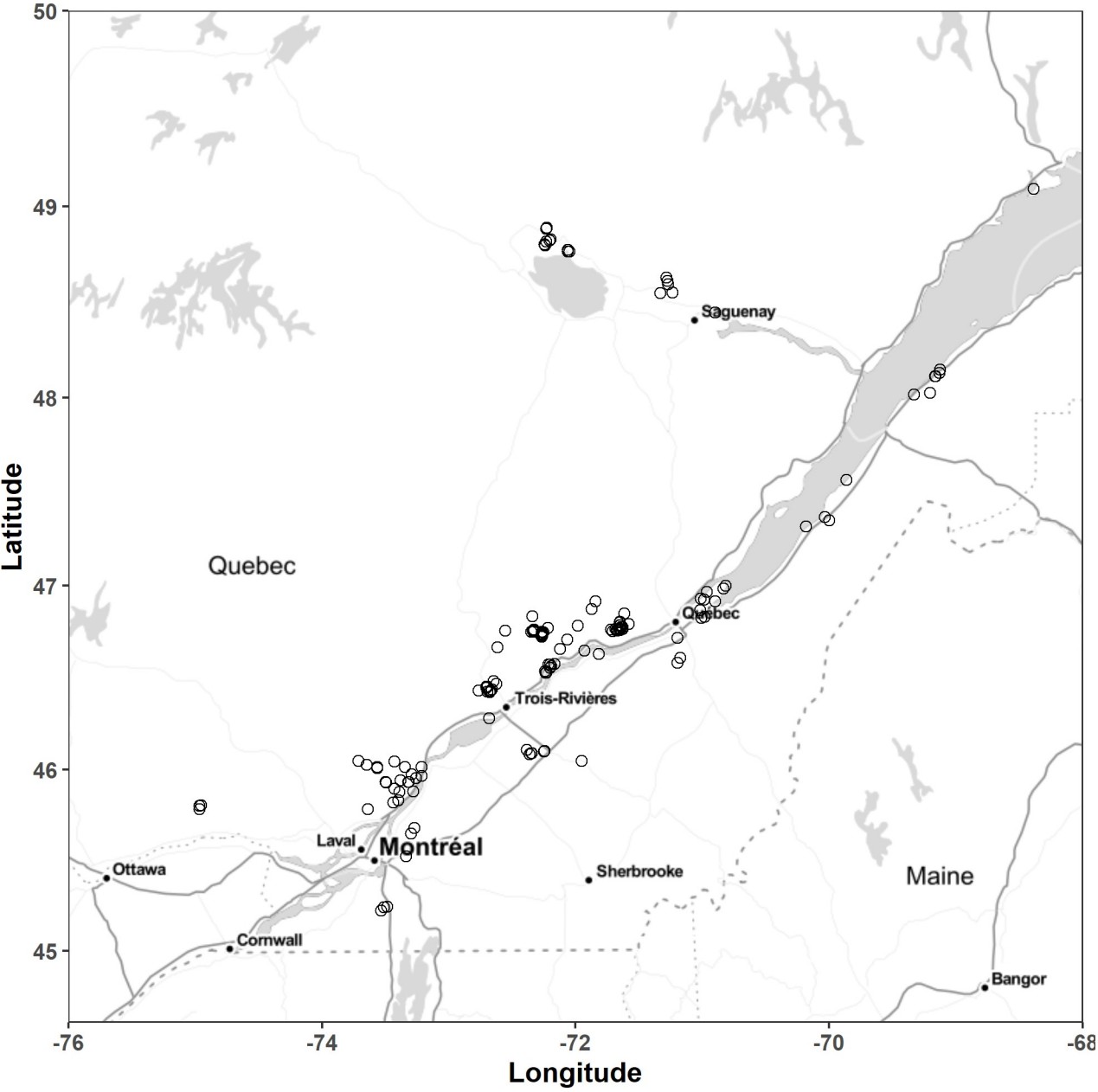

**Fig 1. Location of experimental sites [53].**

**Table 1. Global structure of the machine learning modeling data sets.**

| A. Marketable yield model dataset | | | |
|---|---|---|---|
| **Period** | **Number of trials** | **Number of samples** | **Percentage (%)** |
| **1979–1980** | 2 | 20 | 0.3 |
| **1981–1990** | 4 | 38 | 0.6 |
| **1991–2000** | 68 | 1768 | 29.9 |
| **2001–2010** | 113 | 2386 | 40.4 |
| **2011–2020** | 86 | 1701 | 28.8 |
| **Total** | 273 | 5913 | 100.0 |
| **B. Tuber-size balances model dataset** | | | |
| **Period** | **Number of trials** | **Number of samples** | **Percentage (%)** |
| **1971–1980** | 0 | 0 | 0.0 |
| **1981–1990** | 0 | 0 | 0.0 |
| **1991–2000** | 44 | 1196 | 26.2 |
| **2001–2010** | 81 | 1703 | 37.4 |
| **2011–2020** | 83 | 1658 | 36.4 |
| **Total** | 208 | 4557 | 100.0 |
| **C. Specific gravity model dataset** | | | |
| **Period** | **Number of trials** | **Number of samples** | **Percentage (%)** |
| **1971–1980** | 0 | 0 | 0.0 |
| **1981–1990** | 0 | 0 | 0.0 |
| **1991–2000** | 61 | 1474 | 34.6 |
| **2001–2010** | 70 | 1144 | 26.9 |
| **2011–2020** | 83 | 1636 | 38.5 |
| **Total** | 214 | 4254 | 100.0 |

found in the target variable. Most experiments have been carried out from 1991 (Table 1). The number of trials, the number of samples, minimum and maximum number of blocks and treatments are given in S1 and S2 Tables according to the study year and the fertilizer tested.

There were 48 cultivars classified as early (65–70 days), early mid-season (70–90 days), mid-season (90–110 days), mid-season late (110–130) or late maturity (130 days and more) as suggested on the website of the Canadian Food Inspection Agency [54], with 4%, 13%, 62%, 12% and 9% of the samples respectively. The growing season lengths were provided by scouting teams covering the period from seeding to harvest. The names of the cultivar maturity classes consigned in the data set do not strictly match those of the Canadian Food Inspection Agency [54]. The preceding crop was categorized as in Parent et al. [7] as grasslands, legumes, cereals, low-residue crops and high-residue crops (S3 Table). The data set also includes fertilizers other than N, P or K (classified as NA), fertilizer dosage and application method, seeding density and date, harvest date, tuber marketable yield (excluding tubers < 2.5 cm in diameter), tuber size distribution (small, medium, large) and specific gravity.

## 2.2 Experimental procedures

The experiments included four to six treatments arranged mostly in a randomized complete block design with a minimum of three replications of each treatment (S1 Table). One trial conducted in 1987 had two replications and 8% to 10% of the experiments were arranged as factorial design combining N, P and K fertilizers. We also retained one trial were N, P and K were fixed at their grower-optimum level (S2 Table). Each experimental unit consisted of four or six rows measuring 6 or 8 m in length, with an average row spacing of 0.915 m and within-row

spacing varying with cultivar. The potato seeds were planted in May (excepting June in the Outaouais region) then harvested in September. Median plant density was 36000 plants ha$^{-1}$ in N trials, 33100 plants ha$^{-1}$ in P trials, 36400 plants ha$^{-1}$ in K trials, and 43700 plants ha$^{-1}$ in factorial NPK trials. The N doses varied from 0 to 260 kg N ha$^{-1}$ with varying steps, and P was applied at a dosage of 0 to 130 kg P ha$^{-1}$ with varying steps. The K was applied at a dosage of 0 to 350 kg K ha$^{-1}$ with varying steps. The P and K fertilizers could be converted to $P_2O_5$ and $K_2O$ by multiplying P by 2.291 and K by 1.205. Nitrogen fertilizers were either entirely applied at planting or split-applied between planting and hilling. Phosphorus fertilizers were banded at planting. Potassium fertilizers were band-applied or split-applied before planting and at planting. No animal manure or compost had been applied in the spring and the preceding fall. Other practices were managed uniformly by the grower.

At harvest, 3-m-long ridges in the middle two rows of each plot were dug and hand harvested. Tubers were divided into four categories as follows: culls, small (S), medium (M) or large (L), depending on the smallest diameter size measured with a ruler. The size cut-offs varied with cultivars and market. The marketable yield was calculated as total yield minus culls (tubers < 25 mm in size). Tubers with external defects such as secondary growth and soft rot were discarded. A representative sample of 20 medium-size tubers from each plot was used to determine tuber specific gravity.

## 2.3 Soil characteristics

**2.3.1 Basic soil composition.** Composite soil samples from the 0–20 cm layer were collected in the spring of the study year before planting to determine the initial soil physicochemical characteristics. Particle size distributions were measured as % clay (0–0.002 mm), % silt (0.002–0.05 mm), and % sand (0.05–2 mm) by sedimentation [55] or laser diffraction [56]. Where soil textural classes were not recorded, central values computed for sand, silt, and clay percentages (S4 Table) using the Quebec soil data set [57] were assigned as proxies.

Soil carbon concentration was determined using the Walkley-Black method [58] or Dumas combustion (Leco Instrument, Saint-Louis, MO). The two methods are closely related as in Eq 1 [59]:

$$Dumas\ C(\%) = 0.126 + 1.25 * Walkley-Black\ C(\%) \tag{1}$$

Because soil particle-size distribution and organic matter content are compositional data, they were transformed into isometric log-ratios (ilr) to avoid self-redundancy, non-normal distribution and scale dependency [60]. The ilr transformation consists in log ratios of the geometric means of hierarchically-arranged components and groups of components, and can be interpreted as balances [61]. The hierarchical arrangement of components follows a balance scheme where balances split groups of components sequentially until each group contains a single part. Each balance is computed as in Eq 2:

$$ilr_j = \sqrt{\frac{r_j s_j}{r_j + s_j}} ln\left(\frac{g(c_j^+)}{g(c_j^-)}\right) \tag{2}$$

where for the j$^{th}$ balance in [1,..., D-1] (D is the length of the compositional vector), $r_j$ is the number of parts on the left-hand side, $s_j$ is the number of parts on the right-hand side, $c_j^-$ is the compositional vector at the left-hand side, $c_j^+$ is the compositional vector at the right-hand side, and g() is the geometric mean function. Hence, the textural components and carbon content were balanced as [Sand, Silt, Clay | C], [Clay | Sand, Silt] and [Silt | Sand]. We followed the [denominator parts | numerator parts] notation [62].

**2.3.2 Soil pH.** Soil pH was measured in water (1:1, v/v) or in a 0.01 M CaCl$_2$ solution (1:1 v/v) [63]. The pH$_{CaCl2}$ was converted into pH$_{H2O}$ where required, as in Eq 3 [64]:

$$pH_{water} = 0.27 + 1.03 \, pH_{CaCl2} \qquad (3)$$

**2.3.3 Soil Mehlich-3 extractable P, K, Al, Mg and Ca.** Soil P was extracted using the Mehlich-3 method [65] or Bray-2 converted to P Mehlich-3 values using the Khiari et al. [43] equation as in Eq 4:

$$P_{Mehlich-3}(mg \, kg^{-1}) = -34.6 + 0.86 * P_{Bray-2}(mg \, kg^{-1}) \qquad (4)$$

Soil Al was extracted using the Mehlich-3 method or, where not available, from the typical Al-Mehlich-3 value of soil series as reported by Tabi et al. [57]. Soil K, Ca and Mg were extracted using the ammonium acetate method or its closely-related Mehlich-3 extractant [66]. The P concentration was determined colorimetrically [67] or by inductively coupled plasma (ICP). The K concentration was determined by flame emission or ICP, and Ca, Mg, and Al concentrations were quantified by atomic absorption spectrometry or ICP.

Soil chemical compositions were partitioned into two simplexes $S$(P, Al) and $S$(K, Ca, Mg). The ilr variables were [Fv | Al, P], [Al | P] on the one hand and [Fv, Mg, Ca | K], [Fv | Mg, Ca], [Mg | Ca] on the other.

**2.3.4 Soil profiles.** The soils in our data set were classified according to the Canadian Soil Classification Working Group [68] and ordered along a gleyzation-podzolization gradient using tools of pedometrics [69]. Soil profile reflects the influence of subsoil on crop growth, in particular its impact in regulating the availability of water [70]. The continuous expressions for Quebec potato soil types defined by Leblanc et al. [69] and used by Parent et al. [7] *i.e.*, poorly-drained loam, poorly-drained sand and well-drained sand, were balanced as [Gleyed | Podzol-ized] and [Loamy gleyed | Sandy gleyed].

## 2.4 Weather data

Weather data were collected from the Environment Canada information system [71] using geographical coordinates for each site. The selected weather indexes were the cumulative precipitation–PPT, the Shannon Diversity Index for rainfall distribution–SDI [72], the mean temperature, and the number of growing degree days–GDD.

The cumulative precipitation was computed as the sum of daily rainfall from planting to harvest. The Shannon Diversity Index is the precipitation evenness or the fraction of daily rainfall relative to the total rainfall in a given time period (in days). A SDI = 1 implies complete evenness *i.e.*, equal amounts of rainfall in each day of the period while a SDI = 0 implies complete unevenness *i.e.*, all rain in 1 day [72]. The mean temperature was computed from the planting date to harvest date. The growing degree days index was computed using daily mean temperatures and using 5°C as baseline temperature (*i.e.*, sum of daily mean temperatures equal or superior to 5°C only). Weather variables were computed as in (Table 2) for the period

**Table 2. Equations to compute climatic indices.**

| Index | Description | Unit | Formula | |
|-------|-------------|------|---------|---|
| PPT | Cumulative precipitations | mm | $PPT = \sum_{i=1}^{n} Rd_i$ | |
| SDI | Shannon Diversity Index for rainfall | Unitless | $SDI = \frac{-\sum_{i=1}^{n}[P_i ln(P_i)]}{ln(n)}$ | $P_i = \frac{Rd_i}{PPT}$ |
| T | Mean temperature | °C | $\frac{1}{n}\sum_{i=1}^{n} Tm_i$ | |
| GDD | Growing degree-days | °C | $GDD = \sum_{i=1}^{n} Tm_i \text{ with } Tm_i \geq 5$ | |

Rd is daily rainfall, n is the number of days and Tm is daily mean temperature.

between planting and harvest dates using the historical weather data of the past 5 years (from the corresponding study year) at each site.

## 2.5 Selection of features

**2.5.1 Predictive features.** The study focused on potato yield-impacting factors reported by Parent et al. [7]. Candidate variables were soil Mehlich-3 P, K, Mg, Ca, Al and Fe composition, soil pH, and soil profile classes expressed as balances across soil textural gradients and across gleization-podzolization processes as in Leblanc et al. [69]. The length of the growing season, the preceding crop categories, seeding density and N, P and K fertilizer dosages were used as land management variables. The average 5-yr temperature (T), PTT, GDD and SDI were used as weather features.

The importance of features can be assessed by assigning them a score based on how useful they are at predicting a target variable. We assessed features importance using *ExtraTreesRegressor* function from the scikit-learn Python package [73] on the training set of each target variable.

**2.5.2 Target variables.** The data set is a collection of several experiments with specific objectives. Target variables were total yield, yield fractions, and SG. We separated marketable yield fractions with respect to tuber size as follows: large (L), medium (M) or small (S) size. Because these three fractions must add up to 100% of the marketable yield, they were treated as compositions. These compositional variables were transformed into isometric log-ratios of large-size tubers divided by the geometric mean of small- and medium-size tubers [M, S | L], and medium-size tubers divided by small-size tubers [S | M]. Since analysis of compositional data based on log-ratios of parts is not suitable when zeros are present in a data set [74], we proceeded by firstly imputing zero observations [75], reported mostly for large-size tubers. The detection limit was fixed at 65%. Table 3 summarizes the variables used for modeling. Tuber SG was determined by the weight-in-air to weight-in-water method [76] as in Eq 5:

$$SG = \frac{Weight\ in\ air}{Weight\ in\ air\ minus\ Weight\ in\ water} \tag{5}$$

## 2.6 Data preprocessing

The data were partially preprocessed in the R 3.6.2 statistical computing environment [77]. The tidyverse 1.3.0 package [78] was used for general data handling and visualization. The compositions 1.40–3 package [79] functions helped to transform compositional data into isometric log-ratios, and the robCompositions 2.2.0 package [80] helped to robustly impute missing values. The replacement of zeros in tuber sizes was performed using zCompositions 1.3.3–1 package [75].

The data preprocessing continued in Python 3.8.1 software [81]. The data set used to model tuber SG was cleaned of outliers using the Python SciPy package version 1.4.1 [82]. We used a z-score *i.e.*, a signed number of standard deviations by which the value of an observation or data point is above the mean value of what is being measured on the multivariate data set. The threshold of the score value was set at 3. The data were handled in Python using NumPy version 1.17.5 [83] and pandas 1.0.0 [84] libraries. The matplotlib 3.1.3 package [85] was used for data visualization.

All the quantitative variables were scaled and centered to obtain zero mean and unit variance. The categorical variables were encoded by declining their factors in binary columns, each of which was denoted by 1 to specify the membership of the group of the column, and 0 otherwise.

**Table 3. Variables used for modeling.**

**A. Predictive variables**

| Variable | Type | Description |
|---|---|---|
| **N, P, K doses** | Numeric (kg ha$^{-1}$) | Fertilizer doses used during the experiments |
| **Planting density** | Numeric (plants ha$^{-1}$) | The number of plants within 1 ha of area |
| **Preceding crops** | Categorical | Crop existing on the experimental site along the previous season categorized as small grain, high-residue crop, legume, grassland and low-residue crop (S5 Table) |
| **Growing season length** | Numeric (days) | Number of days between planting and harvest |
| **Temperature** | Numeric (˚C) | Average daily mean temperature from planting to harvest (Table 2) computed with temperature data through the five seasons preceding the season of the study |
| **Precipitations** | Numeric (mm) | Sum of daily rainfall from planting to harvest (Table 2) computed with data through the five seasons preceding the season of the study |
| **Shannon diversity index** | Numeric (unitless) | Precipitations evenness from planting to harvest (Table 2) computed with data through the five seasons preceding the season of the study |
| **Number of growing degree days** | Numeric (˚C) | Sum of daily mean temperature from planting to harvest (Table 2) computed with temperature data through the five seasons preceding the season of the study (5˚C as baseline) |
| **Soil texture (0–20 cm) and carbon** | Numeric (unitless) | Ilr coordinates: [Sand, Silt, Clay | C], [Clay | Sand, Silt] and [Silt | Sand] |
| **Soil types** | Numeric (unitless) | Ilr coordinates representing drainage capacity: [Gleyed | Podzolized] and [Loamy gleyed | Sandy gleyed] |
| **Soil pH** | Numeric (unitless) | Soil pH measured in water or expressed as pH in water |
| **Soil chemical composition** | Numeric (unitless) | P and its fixation agent Al, ilr coordinates: [Fv | Al, P], [Al | P] |
| | Numeric (unitless) | K, Ca and Mg, ilr coordinates: [Fv, Mg, Ca | K], [Fv | Mg, Ca], [Mg | Ca] |

**B. Target variables**

| Variable | Type | Description |
|---|---|---|
| **Marketable yield** | Numeric (Mg ha$^{-1}$), 1Mg = 1000 kg | Sum of small-, medium- and large-size tuber weight |
| **Yield ratios** | Numeric (unitless) | Ilr coordinates of large-size against small- and medium-size tuber weight [M, S | L], medium-size tubers against small-size tuber weight [S | M] |
| **Tuber specific gravity** | Numeric (unitless) | Ratio of tuber weight-in-air to weight-in-water (Eq 5) |

## 2.7 Training and testing data sets

Schemes for partitioning data into training and testing sets vary between studies. Fortin et al. [6] used 60% for training and 40% for testing. Parizeau [86] suggested 50%, 20% and 30% for training, validation and testing, respectively. Crisci et al. [87] used a 75%–25% split while Chantre et al. [88] used a 82%–18% partition for training and testing, respectively. In this paper, the corresponding total input/output data pairs were divided into 70% for training and 30% for testing and model accuracy assessment. Soman and Bobbie [89] found shorter learning times and highest accuracies with such split proportions. Moreover, self-contained and representative data collection is an important step to ensure the sufficiency and integrity of the training data [90]. Thereby, we partitioned the data set according to whether the tested element was N, P K, factorial design or another element (Mg, Ca). Thereafter, data were split at block level to avoid testing models on blocks comprising training samples.

## 2.8 Training models

**2.8.1 Machine learning algorithms.** Four machine learning models were trained to derive an optimal model: *k*-nearest neighbors (KNN), random forest (RF), neural networks (NN) and Gaussian processes (GP). Model parameters were tuned using the random search with cross-validation method (*RandomSearchCV*) of the scikit-learn library version v0.22.1 [73].

**2.8.2 Mitscherlich model.** We used a Mitscherlich-related 3D response surface for three variables inspired by Dodds et al. [91] in the multilevel modeling scheme of Parent et al. [7]. The Mitscherlich-related multilevel response surface was used as a predictive model for comparison with machine learning algorithms. The model was trained using the following equation:

$$Y = A \text{ x}(1 - e^{-R_N \text{x}(E_N + dose_N)})\text{x}(1 - e^{-R_P \text{x}(E_P + dose_P)})\text{x}(1 - e^{-R_K \text{x}(E_K + dose_K)}) \tag{6}$$

where $Y$ is the target variable *i.e.*, marketable yield, $A$ (for *Asymptote*) is the value of the target variable toward which the curve converges at increasing dosage, $E$ (for *Environment*) describes the fertilizer-equivalent N ($E_N$), P ($E_P$) and K ($E_K$) doses from the environment, and $R$ *(Rate)* is the steepness of the curve relating each fertilizer equivalent environmental supply to *Asymptote*. The first-level parameters ($A$, $E$ and $R$) were modeled as linear combinations of the predictors with random effect added to the intercept of the *Asymptote*. To make comparison with preceding models, the model performances were computed without any random effect (*level = 0*). The Mitscherlich multilevel model was fitted in R 3.6.2.

## 2.9 Evaluation of model performance

In all cases, the goodness-of-fit measure or predictive capacity of the developed models was based on the coefficient of determination ($R^2$), the mean absolute error (MAE) and the root-mean-square error (RMSE). The $R^2$ evaluates the proportion of variance in the target variable explained by the model as in Eq 7:

$$R^2 = 1 - \frac{\sum_{i=1}^{n}(y_i - \hat{y}_i)^2}{\sum_{i=1}^{n}(y_i - \bar{y}_i)^2} \tag{7}$$

where $y_i$ is the observed target variable value, $\hat{y}_i$ is the predicted target variable value, and $\bar{y}_i$ is the mean of observed target variable. The best possible score of $R^2$ is 1 (or 100%), but the score may also be negative when the model is arbitrarily worse. Higher $R^2$ values indicate less error variance. A constant model that always predicts the expected value of y disregarding the input features would yield a $R^2$ score of 0 [73]. Typically, values greater than 0.5 are considered acceptable [92]. The MAE is the average of the absolute differences between predictions and observations as in Eq 8:

$$MAE = \frac{1}{n}\sum_{i=1}^{n}|y_i - \hat{y}_i|. \tag{8}$$

The MAE attributes equal weight to individual errors and is less sensitive than $R^2$ or RMSE to large prediction errors. The RMSE is the square root of the average of squared differences between predictions and observations computed as in Eq 9:

$$RMSE = \sqrt{\frac{1}{n}\sum_{i=1}^{n}(y_i - \hat{y}_i)^2}. \tag{9}$$

The RMSE attributes high weight to large errors due to squaring. Both MAE and RMSE indicate prediction errors in the units of variable of interest. Zero values indicate a perfect fit.

Values less than half of the standard deviation of measured data were considered low [93]. The trained models were used to predict optimal N, P and K doses using some left-out experimental sites data.

### Economic or agronomic optimal doses

The optimal nutrient input is the one returning yield of high-quality tubers [32], where profitability is maximized and the environmental footprint minimized [94, 95]. To compute the optimal economic N, P, K doses at a given site, all the predictive features, but not N, P and K doses, were held constant (fixed input data). The row of fixed input variables is stacked (reproduced) 1000 times to obtain a table with 1000 identical rows. We generated 1000 random N-P-K combinations of doses from uniform distributions of plausible doses varying between zero and 250 kg ha$^{-1}$ for N, 110 kg ha$^{-1}$ for P, 208 kg ha$^{-1}$ for K. The table was altered in such a way that only N-P-K dosage changed following the random combinations.

A fertilizer cost was computed for each N-P-K triplet. Unit fertilizer costs were set at $1.20 CDN kg$^{-1}$ for N, $1.10 CDN kg$^{-1}$ for P and $0.90 CDN kg$^{-1}$ for K. Tuber price was set at $250 CDN Mg$^{-1}$ (1 Mg = 1000 kg) as in Parent et al. [7]. No environmental footprint effect was used because of a lack of reliable sources, although they could have been implemented as an increase in the cost of unit dosage. The difference between fertilizer cost and tuber revenue provided the marginal benefit from fertilizing. Economic optimal N-P-K dosage was reached where the net return was maximum. For tuber size and SG, an agronomic optimal N-P-K fertilizer dosage was deducted where the target variable reached a maximum.

Our results are reproducible by using the codes, data and package requirements provided in a GitHub repository at https://git.io/JvYxd.

### 2.11 Model interpretation data

We randomly selected four trials in the testing set for model interpretation (Table 4). The trials showed soil pH levels ranging between the adequate limits of 5.2 to 6.2 for potato crops according to the *Centre de Référence en Agriculture et Agroalimentaire du Québec* [96]. The

**Table 4. Description of trials used for model analysis.**

|  | Trial 194[*] | Trial 8804 | Trial 412 | Trial 320 |
|---|---|---|---|---|
| **Nutrient tested** | P | N | P | K |
| **Cultivar** | Superior | FL 1533 | Goldrush | Krantz |
| **Maturity class** | Early mid-season | Mid-season | Mid-season | Mid-season |
| **Growing season length (days)** | 102 | 131 | 108 | 112 |
| **Planting density (seeds ha$^{-1}$)** | 36430 | 43716 | 36433 | 31226 |
| **Mean temperature (5 years) T°C** | 16 | 18 | 16 | 18 |
| **Total rainfall (5 years) mm** | 378 | 359 | 363 | 448 |
| **Soil pH** | 5.5 | 5.5 | 5.8 | 6.1 |
| **Soil P (Mehlich 3) mg kg$^{-1}$** | 23 | 175 | 46 | 349 |
| **Soil K (Mehlich 3) mg kg$^{-1}$** | 83 | 265 | 72 | 200 |
| **Soil Al (Mehlich 3) mg kg$^{-1}$** | 1580 | 1570 | 2839 | 1216 |
| **ISP$_1$ (environmental index %)** | 1.4 | 11.1 | 1.6 | 28.7 |
| **Texture** | Sandy loam | Fine sand | Sandy loam | Loam |
| **Minimum dose (kg ha$^{-1}$)** | 0 | 0 | 0 | 0 |
| **Maximum dose (kg ha$^{-1}$)** | 300 | 200 | 200 | 300 |

[*] Trial n° 194 used for economic optimal and agronomic optimal doses computation. Al: aluminium, ISP: phosphorus saturation index

phosphorus saturation environmental index $(P/Al)_{Mehlich3}$ classified the sites at extremely low environmental risk for P trials (1.4% to 1.6%), medium risk for N trial (11.1%) and very high risk for K trial (28.7%). Soil potassium levels showed extremely low (71.5 mg kg$^{-1}$) and very low (83.1 mg kg$^{-1}$) levels for P trials, medium level for K trial and high level for N trial [97].

## 3. Results

**Feature importance.** The feature importance, computed using the *ExtratreesRegressor* function, revealed that the N fertilizer dose was by far the most informative feature in the marketable yield prediction models, followed by soil type, air temperature, length of growing season and soil texture. To predict large-size tuber yield ([M, S | L] balance), the N dose remained the most informative feature, followed by soil type and texture. Tuber planting density exceeded other features for medium-size tubers ([S | M] balance), followed by N dose, soil elements (P and Al Mehlich-3) and soil type. For tuber SG, weather indices, *i.e.*, Shannon diversity index, total rainfall and temperature, returned the highest scores (Fig 2). Preceding crops were not informative across target variables and were deleted before modeling.

### 3.2 Model tuning parameters

The tuning parameters varied within the models depending on target variables (Table 5). The parameters were tuned during modeling using python random search method with 5-fold cross-validation. For each target variable the corresponding training set was used.

The basic assumption in the KNN algorithm is that similar samples should return similar output (class or value) [98]. The two parameters to tune are the distance function which determines the similarity, and the optimal number of neighbors (similar known observations, $k$) to use for assigning the unknown output. The regressions were run with 19 nearest neighbors ($k$ = 19) for yield, tuber size [M, S | L] balance and SG prediction models. For the [S | M] balance prediction model, k was set at 18 neighbors. With uniform weights, all the points in each neighborhood are weighted equally while with an inverse distance weight, closer neighbors have a greater influence than neighbors which are further away.

The parameters of a RF include mainly the number of decision trees in the forest and the number of features considered by each tree when splitting a node. The optimization procedure set the number of trees in the forests to 92, 12, 17 and 19 for yield, tuber size [M, S | L] balance, tuber size [S | M] balance and SG prediction models, respectively. The number of features considered for splitting at each leaf node were selected automatically.

A NN is characterized by its architecture, the training algorithm and the activation function. We used a multilayer perceptron in which neurons are organized in layers: an input layer where data are fed into the system, one or more hidden layers where the learning takes place, and an output layer where the decision/prediction is given [99]. We tuned the number of neurons for one hidden layer, and the activation function. A hyperbolic tangent activation function was selected for all the target variables prediction models. The tuned numbers of the hidden layer neurons were 100, 200, 100 and 200 for yield, tuber size [M, S | L] and [S | M] balances, and tuber SG respectively.

GPs are defined by a mean function m(x), a kernel or covariance function generating the covariance matrix $k(x_i,x_j)$ between pairs of random outputs. A white noise ($\sigma^2$) can optionally be added to the kernel [100]. The Matern kernel without white noise returned the lowest error for each target variables. Different noise levels were found to be optimal: 0.195 for marketable yield prediction model, 0.136 for tuber size [M, S | L] balance, 0.031 for [S | M] balance, and 0.932 for tuber SG. Because all the target variables were scaled and centered, mean functions m(x) were null.

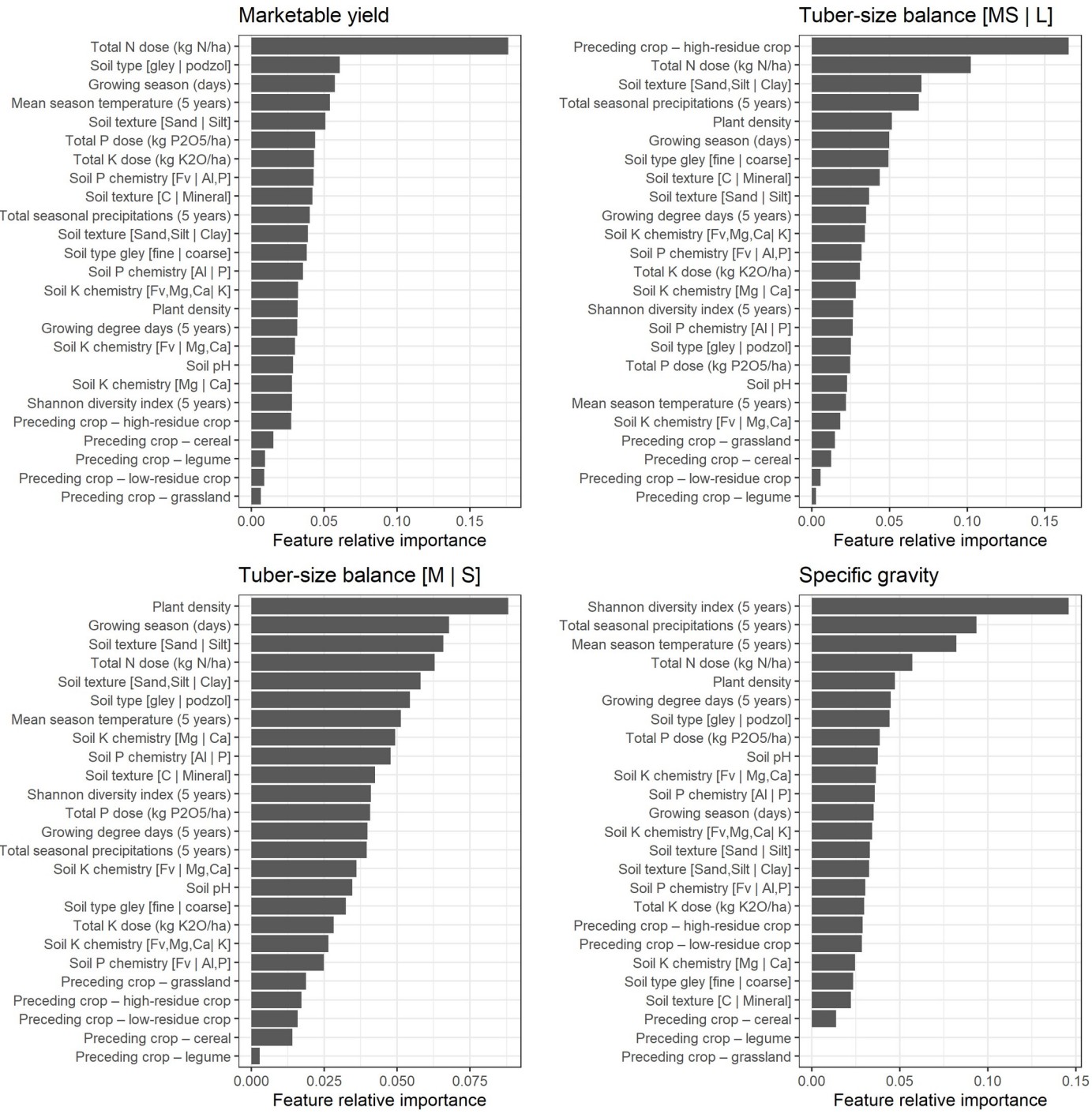

**Fig 2. Predictive features importance for modeling.**

## 3.3 Comparison between models

Model performance to predict marketable yield, tuber-size balances and tuber SG was assessed using $R^2$, MAE, RMSE, response curves shapes and economic optimal N-P-K dosage predictions for each model. For all the models, the predictive accuracy level was not affected after discarding the preceding crop classes.

**Table 5. Tuned model parameters.**

| A. Data used to tune parameters | | | | |
|---|---|---|---|---|
| **Target variable** | **Yield** | **[M, S \| L]** | **[S \| M]** | **SG** |
| **Number of samples** | 5913 | 4557 | 4557 | 3180* |
| **Training set** | 4139 | 3203 | 3203 | 2242 |
| **Testing set** | 1774 | 1354 | 1354 | 938 |
| **B. k-nearest neighbors** | | | | |
| **Target variable** | **Yield** | **[M, S \| L]** | **[S \| M]** | **SG** |
| *K* | 19 | 19 | 17 | 19 |
| **Distance** | Euclidean | Euclidean | Euclidean | Euclidean |
| **Weight** | Inverse distance | Uniform | Inverse distance | Uniform |
| **C. Random forest** | | | | |
| **Target variable** | **Yield** | **[M, S \| L]** | **[S \| M]** | **SG** |
| **Number of trees** | 92 | 12 | 17 | 19 |
| **Number of features** | 'auto' | 'auto' | 'auto' | 'auto' |
| **D. Neural networks** | | | | |
| **Target variable** | **Yield** | **[M, S \| L]** | **[S \| M]** | **SG** |
| **Input layer size** | 20 | 20 | 20 | 20 |
| **Hidden layers size** | 100 | 200 | 100 | 200 |
| **E. Gaussian process** | | | | |
| **Target variable** | Yield | [M, S \| L] | [S \| M] | SG |
| **Kernel** | Matern | Matern | Matern | Matern |
| **Noise level (alpha)** | 0.195 | 0.136 | 0.031 | 0.932 |

\* The total number of samples (3180) differs from that of this target variable in Table 1 because 1074 outliers have been excluded during the process.

**3.3.1 Goodness of fit.** The model scores at training and testing for the different target variables are presented in Fig 3. There was a large gap between training and testing scores. The difference was lower for the Mitscherlich model, which also showed the lowest coefficient of determination and the highest MAE and RMSE. Its $R^2$ values were 0.35 and 0.37 at training and testing, respectively. The $R^2$ values of machine learning algorithm-based models ranged between 0.78 (NN) and 0.92 (KNN) at training, and between 0.49 (NN) and 0.59 (RF) at testing in predicting marketable yield. With the large-size tuber yield balance [M, S | L], the $R^2$ values ranged between 0.72 (KNN) and 0.87 (RF) at training, and between 0.55 (KNN) and 0.64 (GP) at testing. The medium- versus small-size tuber [S | M] balance and SG prediction models were the most informative, as shown by the highest $R^2$ values at both training and testing. The $R^2$ values ranged between 0.83 (NN) and 0.93 (KNN) at training and between 0.62 (RF) and 0.69 (KNN) at testing in predicting small-size tuber balance, while for SG, they ranged between 0.72 (KNN) and 0.94 (RF), then between 0.58 (KNN) and 0.67 (RF) at training and testing, respectively. In general, model MAE and RMSE were slightly higher when $R^2$ values were low. The practically-similar magnitudes between RMSE and MAE meant that all the individual differences between predictions and observations had equal weight.

**3.3.2 Response curves.** The marketable yield response curves are plotted in Fig 4 for each model with respect to the tested nutrient. There were disagreements between models. The Mitscherlich, NN and GP models generated smooth response curves, while the KNN and RF models generated stepped curves. The marketable yield was non-responsive to P application in the RF model. There was also no effect of K fertilization on the yield shown by the Mitscherlich

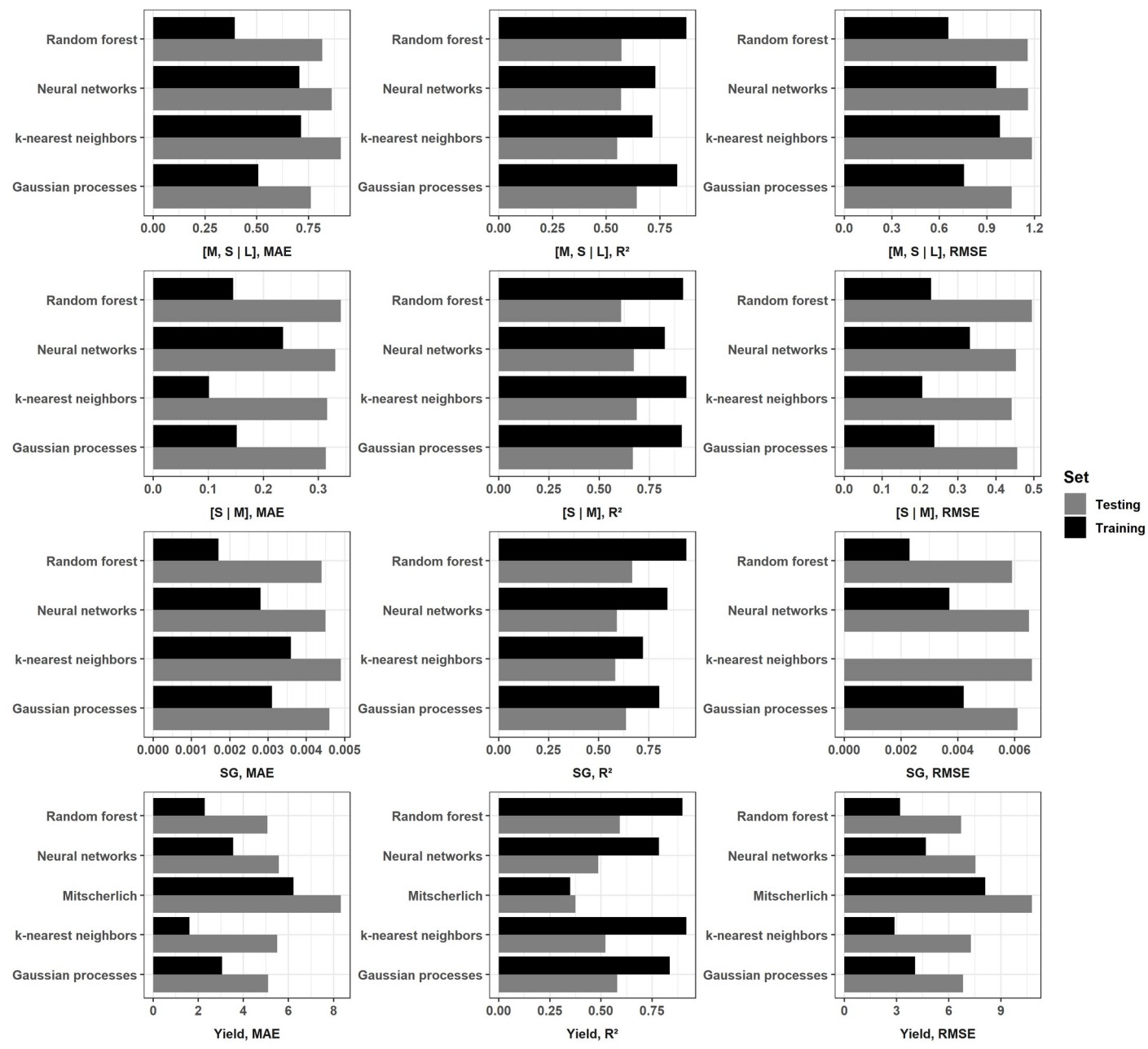

**Fig 3. Comparison of models goodness of fit using R², MAE and RMSE.**

and RF models. All models for the P trial somewhat underestimated marketable yield while response curves followed data for N.

The Mitscherlich model was excluded for the analysis of other target variables. Figs 5–7 show how each model fits responses of tuber size balances ([M, S | L] and [S | M]), and SG, respectively, with respect to N, P or K dosage. The NN and GP models generated smooth curves, while the KNN and RF models generated stepped curves. The [M, S | L] balance (Fig 5) showed increasing response to N fertilization across models, while response was globally poor for P and K. For the [S | M] balance, responses increased with increasing fertilizer doses, except for P and K trials data fitted with GP model (Fig 6). There was also poor response for K trial

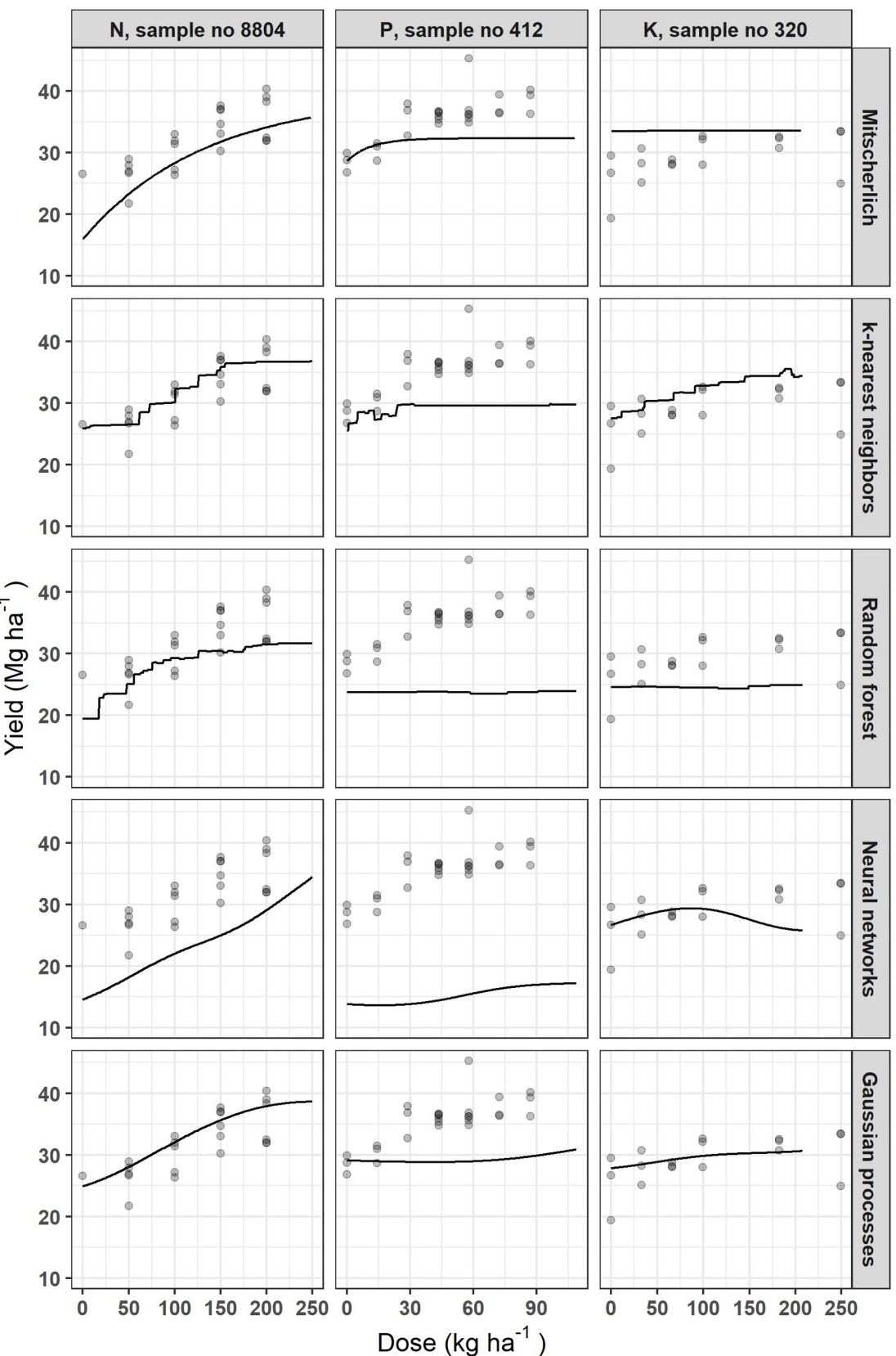

**Fig 4. Examples of potato yield response to N, P or K fertilization using different models.**

with SG (Fig 7). The SG response decreased from zero K levels and increased then decreased as P dosage increased. For N trials, SG slightly increased then decreased as N dose increased in the RF model, but was non-responsive with the other models.

**3.3.3 Predictions.** The fertilizer recommendations and output predictions varied with the model and the target (Fig 8). The Mitscherlich and NN models predicted negligible economic optimal K doses (11 and 12 kg ha$^{-1}$ respectively) in marketable yield prediction models, while the site Mehlich-3 K level was classified as very low (83.1 mg kg$^{-1}$) according to local standards [96]. The RF model suggested the highest cumulative agronomic optimum fertilizer doses, although its outputs were not the highest. With the tuber size [M, S | L] balance prediction model, practicable doses were recommended only by the GP model for P (107 kg ha$^{-1}$) and the RF model for K (185 kg ha$^{-1}$), a scheme that is almost similar to the [S | M] balance prediction models. For this output, the GP model recommended only 17 kg P ha$^{-1}$, while N and K were impracticable (1 kg ha$^{-1}$ and 4 kg ha$^{-1}$, respectively). Despite the extremely low environmental risk for P and the low level of soil K, some models predicted negligible doses of P and K mainly for tuber size balances.

## 3.4 Probabilistic predictions

In addition to point estimates shown by each model, the GP model can return posterior samples. Each sample is a function from which we can compute an economic optimal (marketable yield) or agronomic optimal (size balances or SG) fertilizer dose. Figs 9–12 present the results of 1000 generated samples for each target variable for the selected N, P and K trials. The average GP curve is shown as a black line, with its optimal dosage as a black dot. Five sampled GP curves are plotted as grey lines, with their optimal doses as grey dots. The probability distributions of the 1000 optimal doses are shown under the respective response curves. The figures show that predicted means of optimal dosage (black dot) did not always correspond to the most likely dosage (highest histogram bar) computed after running the sampling process. With yield prediction models (Fig 9), the mean economic optimal dose corresponded to the probabilistic prediction only for the N trial (250 kg N ha$^{-1}$). For the tuber size [M, S | L] balance (Fig 10), the probabilistic prediction was equal to the mean GP prediction for P trial *i.e.*, 87 kg P ha$^{-1}$, while N and K trials returned equal predictions with the [S | M] balance prediction models with 0.0 kg ha$^{-1}$ and 0.70 kg ha$^{-1}$, respectively (Fig 11). For tuber SG prediction models, none of the probabilistic recommendation matched the mean GP optimal dosage (Fig 12).

# 4. Discussion

## 4.1 Selection of features

Fertilization trials were conducted over a time span of four decades (1979–2017). Although agricultural practices, soil conditions and analytical techniques have undergone substantial changes over time, Valkama et al. [101] has shown that the differences between old and recent experiments in yield responses are not statistically important. Moreover, where the analytical techniques for the same element differed, correlation equations were available to converting to one technique before data analysis. It is the case for soil carbon converted from Walkley-Black to Leco CNS (Eq 1), soil pH processed with CaCl$_2$ converted to pH water (Eq 3), and P-Bray-2 converted to P-Mehlich-3 (Eq 4). Since there were similarities in experimental procedures and ability to uniformly convert measurement methods, we found that the data set could be used for machine learning.

The feature selection function selects a subset of variables for a learning algorithm to focus attention on the subset, especially when dealing with a large number of explanatory variables. The model-based approach incorporates the correlation structure between predictors and

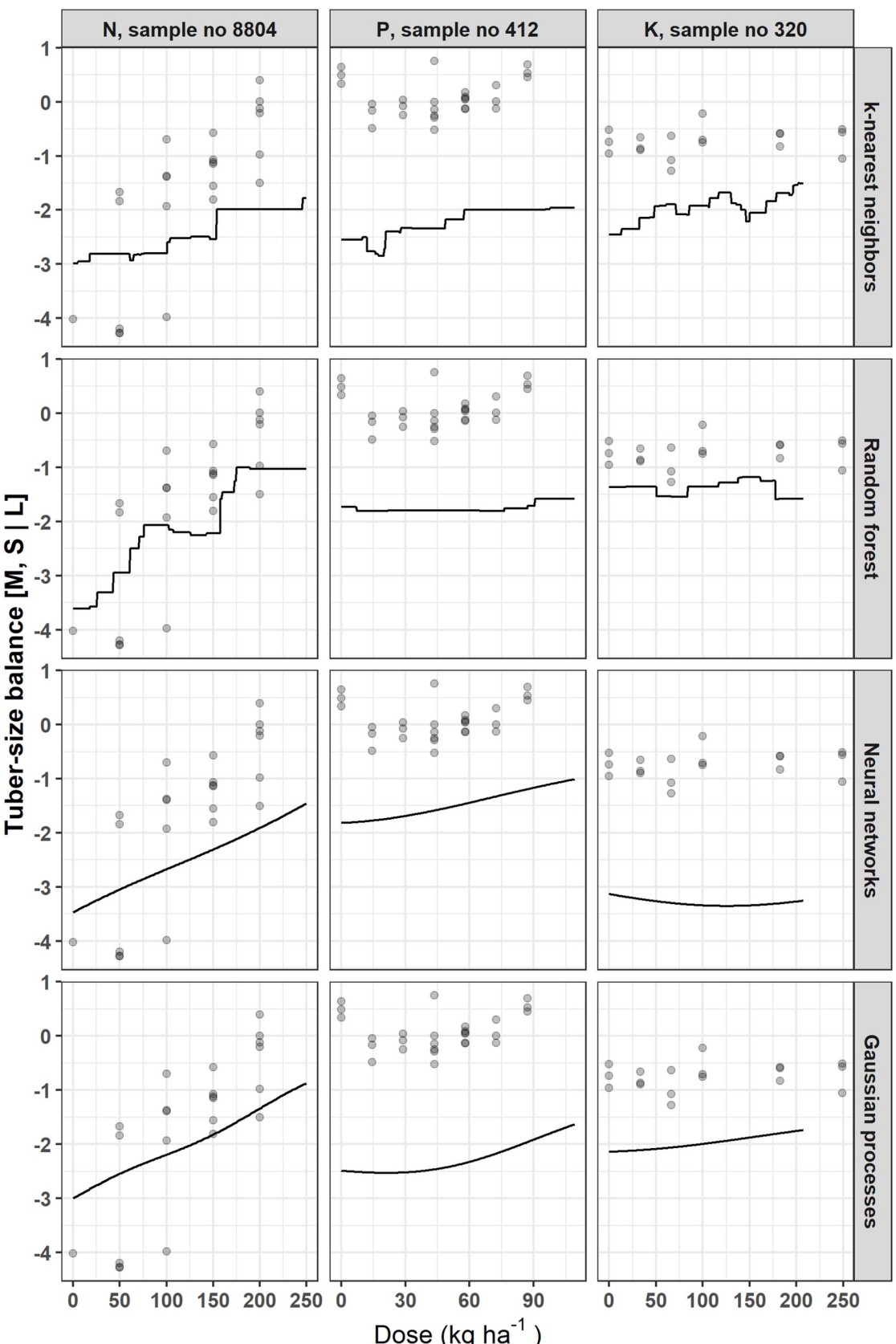

**Fig 5. Examples of potato tuber size [M, S | L] balance response to N, P or K fertilization using different models.**

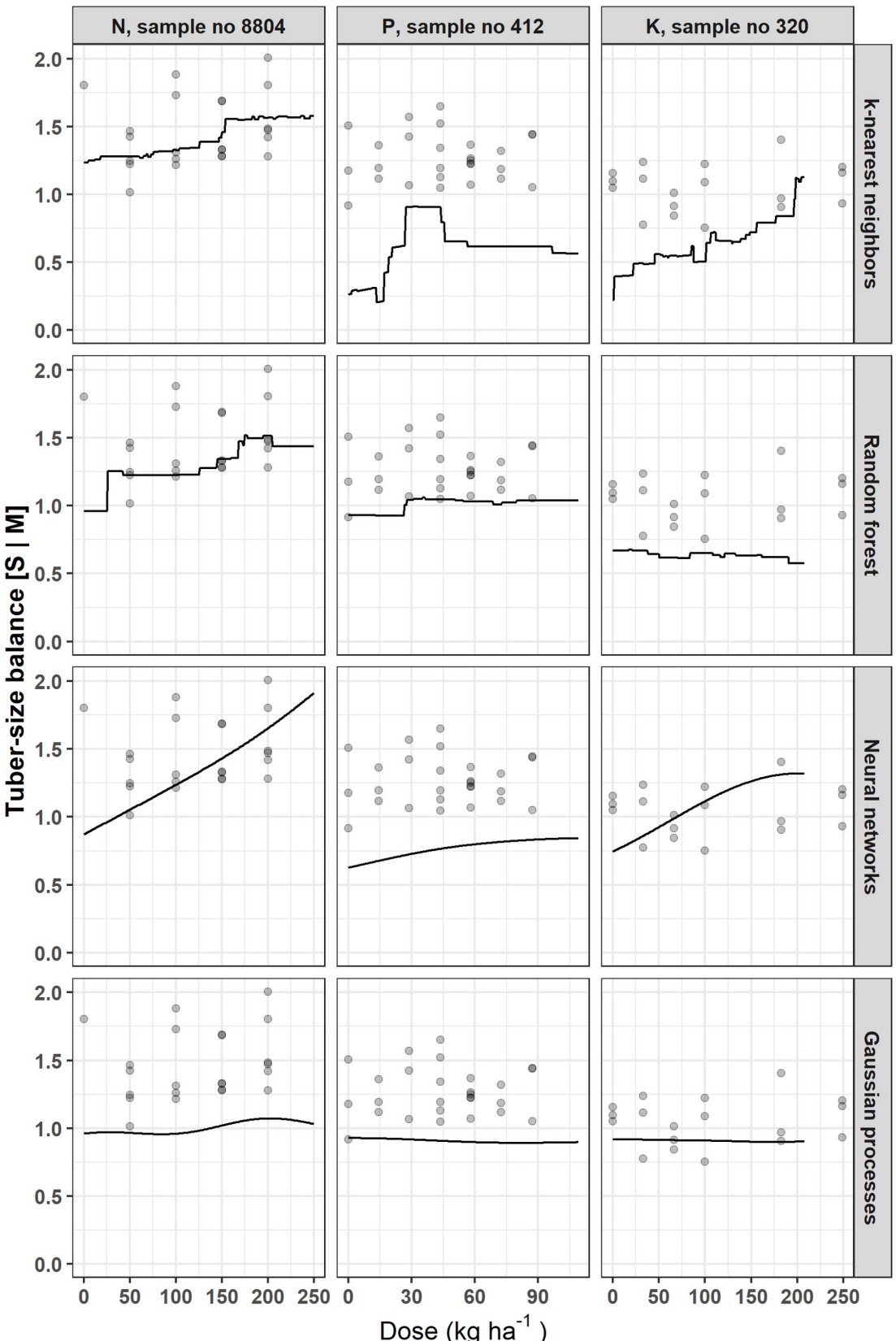

**Fig 6. Examples of potato tuber size [S | M] balance response to N, P or K fertilization using different models.**

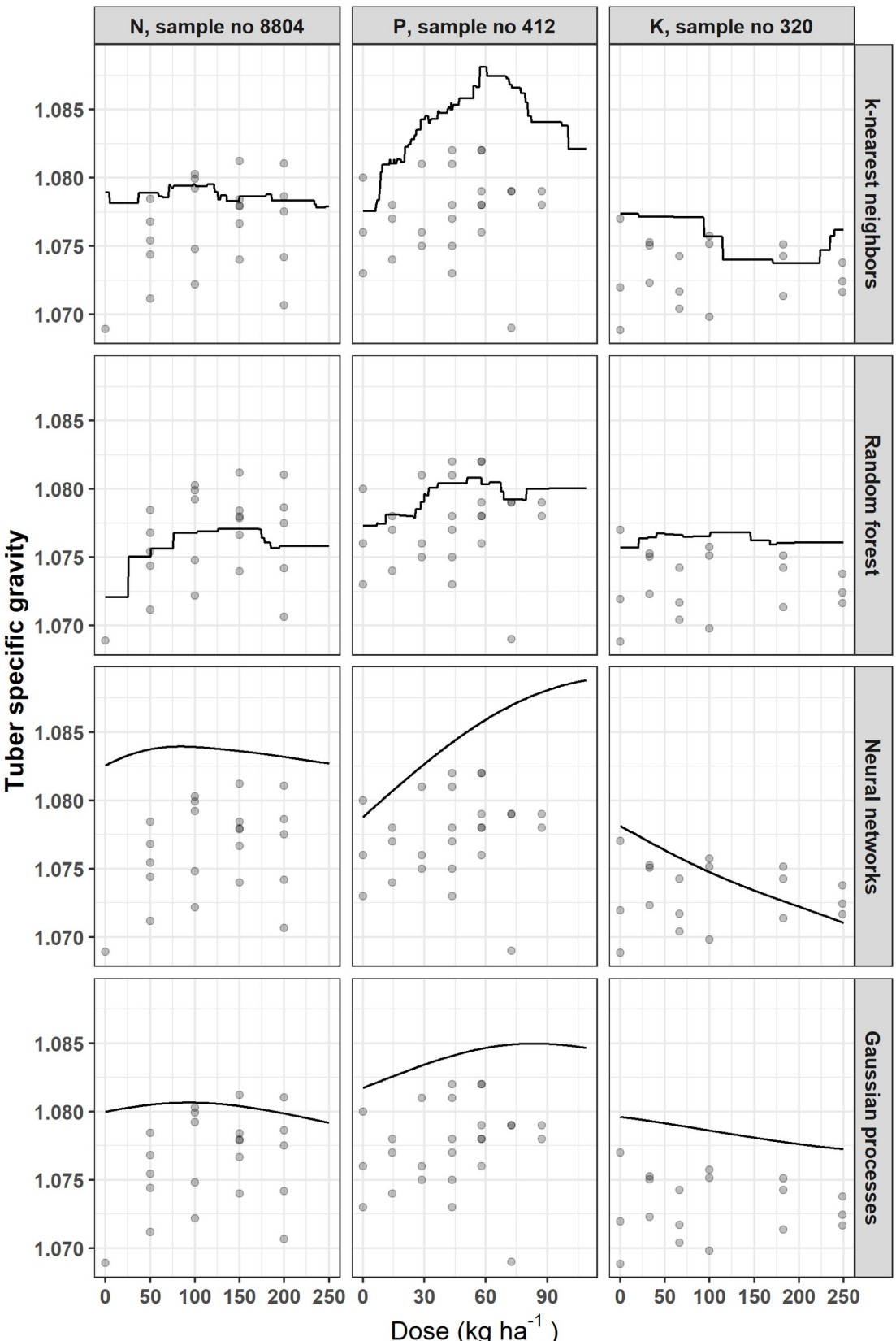

**Fig 7. Examples of potato tuber SG response to N, P or K fertilization using different models.**

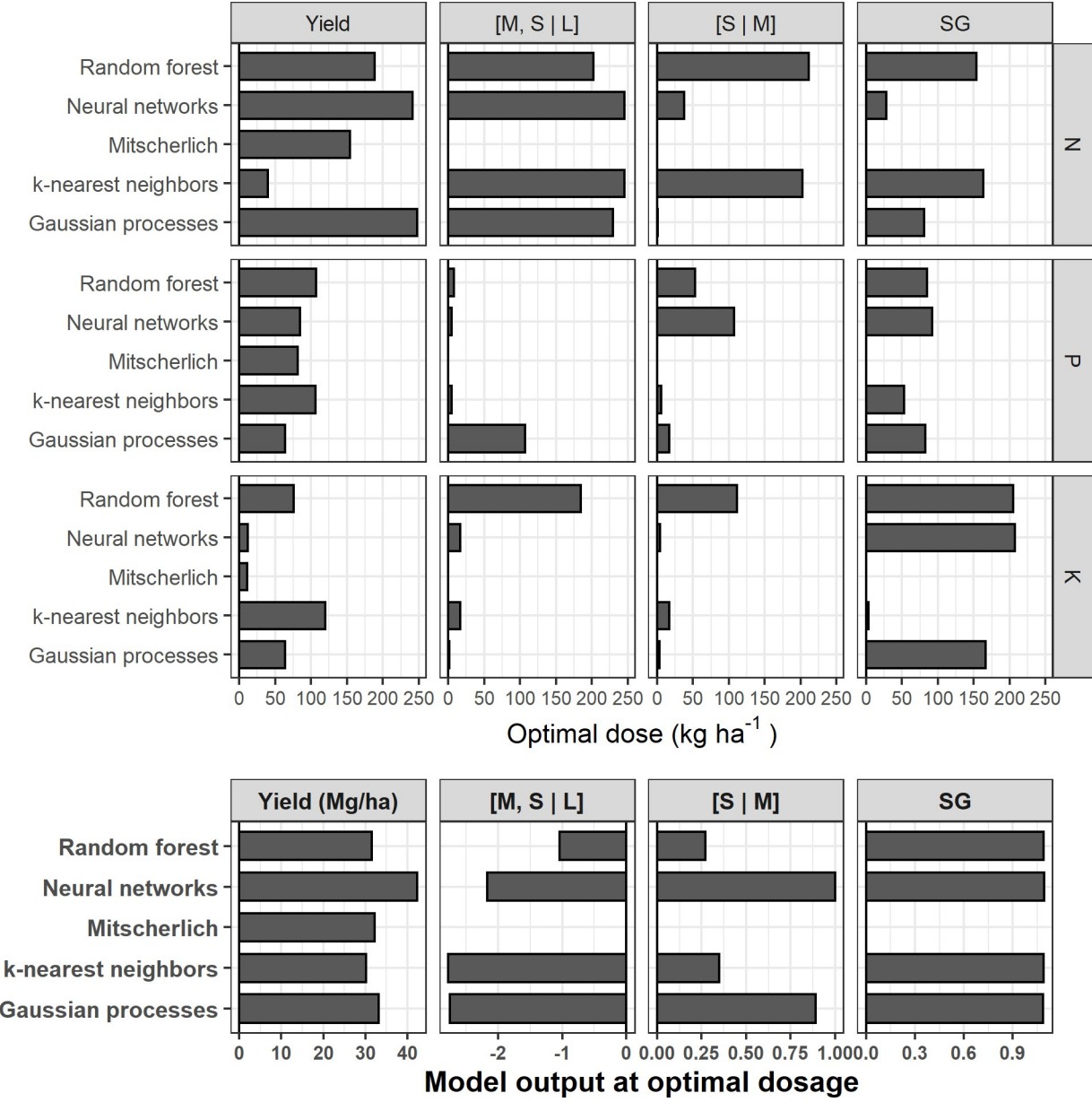

**Fig 8. Economic or agronomic optimal doses and output predictions at optimal dosages for each model with a random selected test trial (N° 194).**

provides scores that indicate how useful or valuable each feature is in model building. Features with low or no importance could be removed without affecting model performance [73]. The preceding crops categories *i.e.*, grassland, small grains, legumes, low-residue crops and high-residue crops, as categorized by Parent et al. [7], returned zero (for tuber SG) or faintest scores (for other target variables) and were thus removed despite a substantial body of literature on the advantages of crop rotation to the next crop. Nonetheless, Zebarth et al. [102] stated that the amount of nitrogen mineralized from organic matter during the growing season cannot be predicted accurately. Torma et al. [103] found that the N supplied by soil and crop residues (maize, potato, silage maize, soybean, sunflower, winter rape, winter wheat) ranged from 20 to

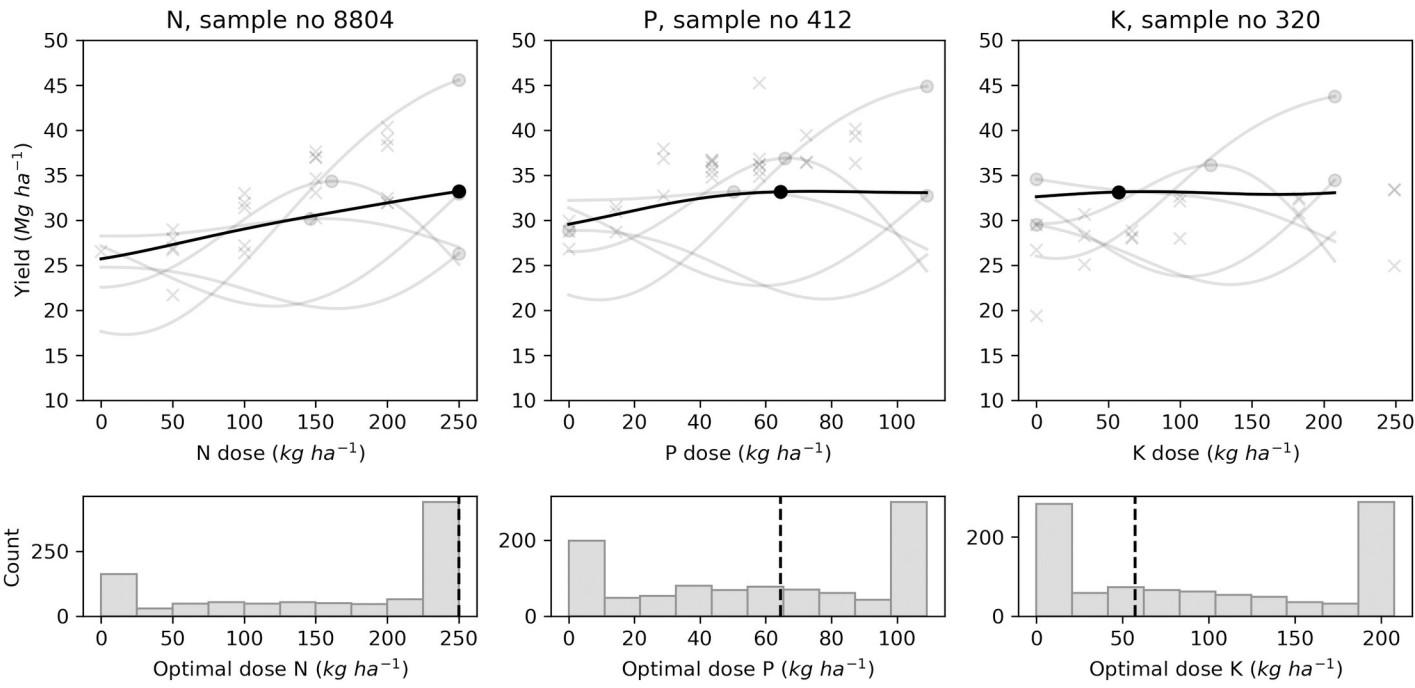

**Fig 9. Examples of optimal economic N, P, K doses distribution with Gaussian processes using marketable yield for selected trials.**

132 kg ha$^{-1}$, while the phosphorus ranged from 2 to 24 kg ha$^{-1}$ and potassium from 13 to 218 kg ha$^{-1}$. Rangarajan [104] stated that nutrient availability to the next crop depends on whether the entire plant or only the root system is left in the field, and on how environmental conditions govern the rate of organic matter decomposition.

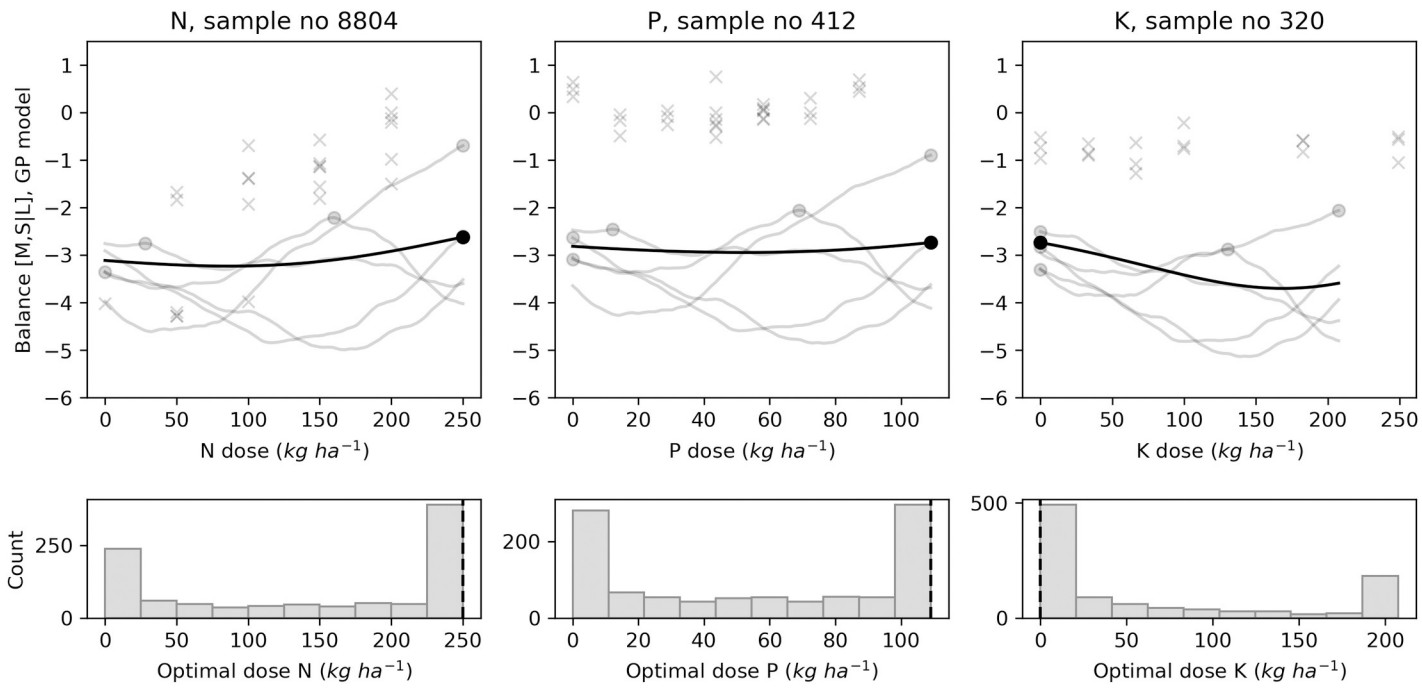

**Fig 10. Examples of agronomic optimal N, P, K doses distribution with Gaussian processes using tuber size [M, S | L] balance for selected trials.**

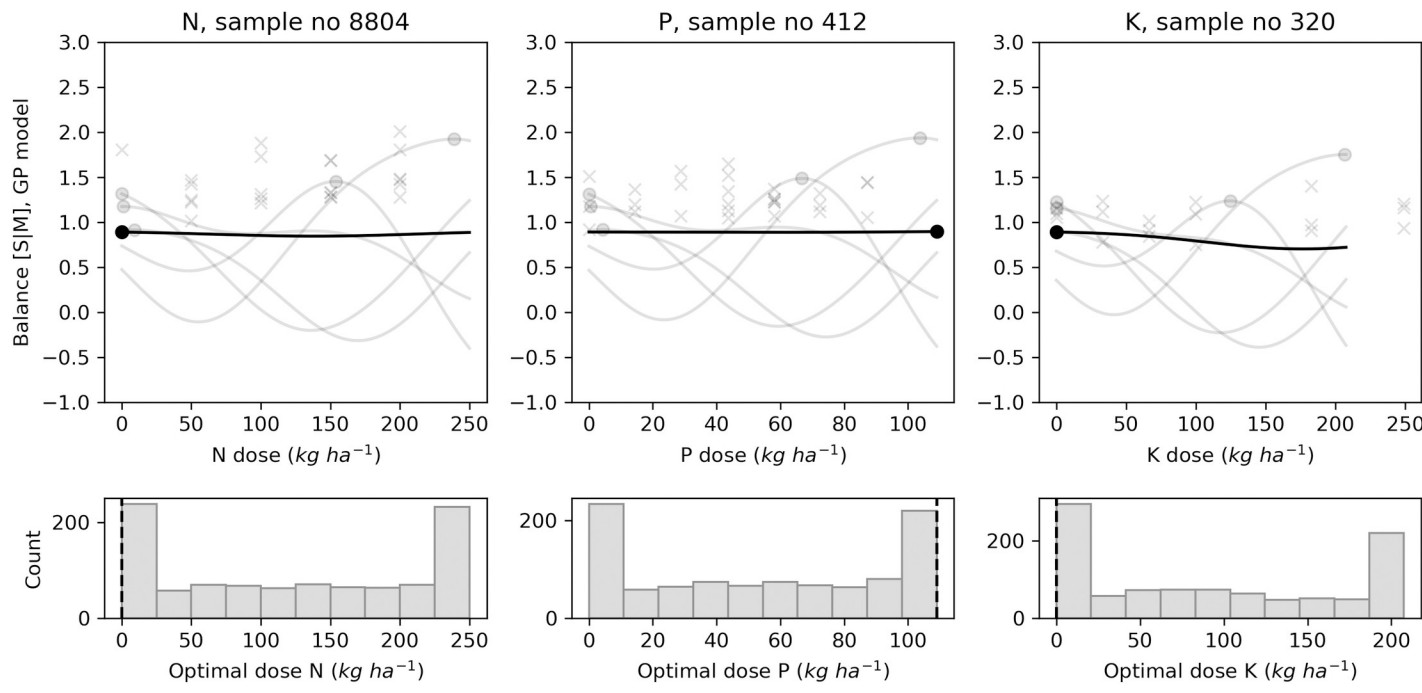

**Fig 11. Examples of agronomic optimal N, P, K doses distribution with Gaussian processes using tuber size [S | M] balance for selected trials.**

For marketable yield and tuber size balances prediction models, the N dose was the most informative feature, probably because of its close relation to photosynthesis [105]. Applied in excess, it delays tuber maturity, stimulates foliage production, increases plant susceptibility to

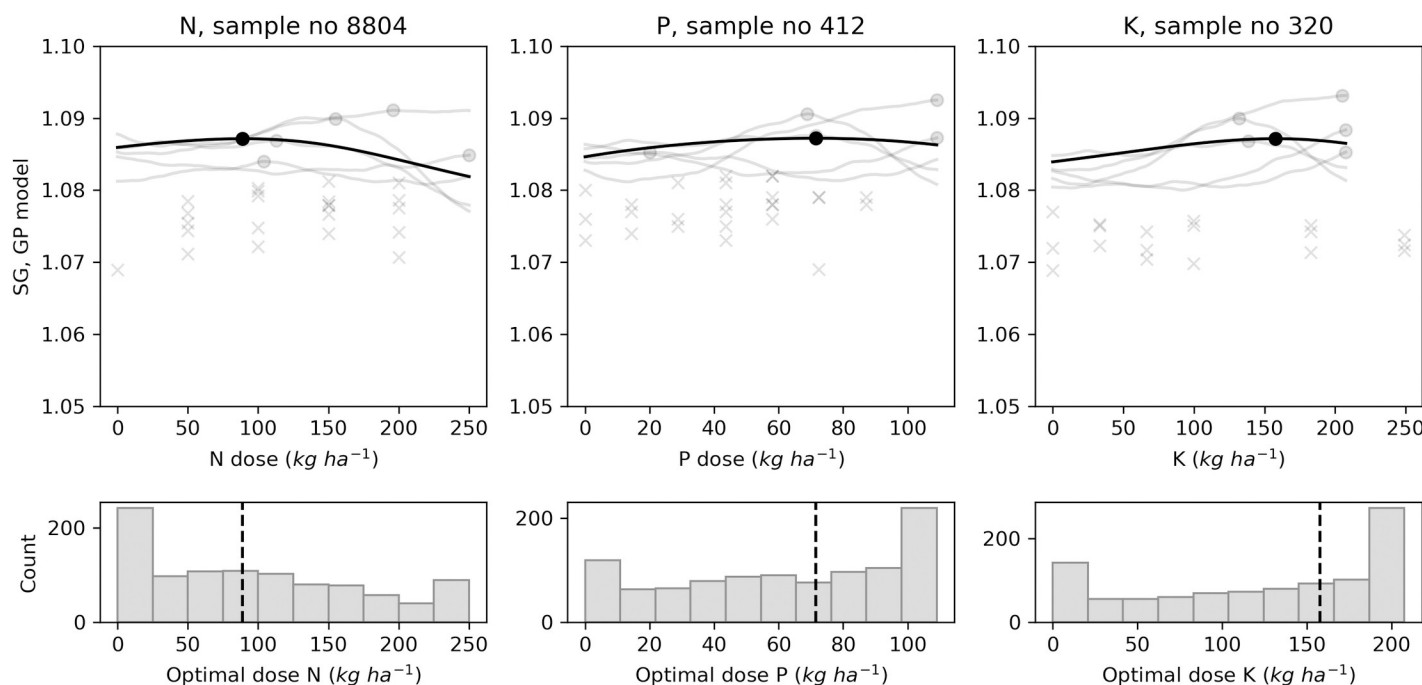

**Fig 12. Examples of agronomic optimal N, P, K doses distribution with Gaussian processes using tuber SG for selected trials.**

diseases and reduces tuber SG [106]. Crop yield is also determined by environmental conditions driving the physical, chemical and biological reactions [107] that are important in empirical or mechanistic models [4, 7–9, 108].

The selection process retained soil profile characteristics and weather events as major features. Levy and Veilleux [109] reported the effects of air and soil temperatures on potato growth mechanisms and tuber yield. Leblanc et al. [69] pointed out soil drainage conditions for loamy-gleyed profiles (poorly-drained loam), sandy-gleyed profiles (poorly-drained sand) and sandy-podzolized profiles (well-drained). Soil compaction has a negative impact on root extension and water movement *i.e.*, the reduction of nutrient uptake potential leading to a severe reduction of tuber yield [8]. Xu et al. [110] developed pedotransfer functions for potato grown on light-textured soils that could be useful in future models.

Dry matter production of potato crops is determined by the length of the growth cycle [111], which turned out to be a valuable feature. Camire et al. [112] stated that long growing season favors high-yielding late-season cultivars. Rex [113] found a close relationship between delayed harvest date and total yield, main-size marketable tubers and SG.

Seeding density was the most informative feature of the medium- to small-size tubers balance. Seeding density differentiates the number of tubers harvested, the weight of the tubers and the size distribution; higher plant densities promote higher yields in small and medium sizes [113–115].

The feature selection algorithm showed the impact of weather indices on tuber SG. The Shannon diversity index, total rainfall and temperature yielded the highest scores in a decreasing order. Al Soboh et al. [116] reviewed the factors affecting SG loss in crops of crisping potato and stressed that irrigation during early growth stages increases tuber dry matter content. Specific gravity could be reduced substantially if heavy rain occurred at the end of the season before harvest. They stated that potatoes grown during a period of increasing day length, temperature and light intensity produce tubers of high SG. In this study, GDD considered only daily mean temperatures higher than or equal to 5˚C as used by Parent et al. [7]. Moulin et al. [117] used a baseline of 7˚C and 30˚C as upper limit. Moreover, the general trend of SG response curves with respect to fertilization supported the results of Belanger et al. [118], Zebarth et al. [19] and Laboski and Kelling [119]. Excessive application doses of N and K along with high soil levels of either nutrient may reduce SG. Phosphorous application may increase tuber solids when soil test P levels are low. Specific gravity was not influenced by the relatively high levels of N and P used by Dubetz and Bole [120], while Maier et al. [121] found contrasted effects between trials.

The relative importance of a variable in a model is related to its effect on the output through its gradient in the data set. Hence the predominance of N doses, and P and K doses to some extent, could have been caused by the origin of the data set, which is a collection of fertilizer trials, where large gradients of doses are found by design. This study did not address fertilizer source and timing of application. While Marouani et al. [122] found equivalency of ammonium nitrate (33.5% N), urea (46% N), NP fertilizer (33% N– 14% $P_2O_5$) and NPK fertilizer (27%N– 5% $P_2O_5$–5% $K_2O$), Petropoulos et al. [123] found that the form of the fertilizer (ammonium sulfate, ammonium sulfate + zeolite, manure, slow release N fertilizer with urease inhibitor) and the cultivar (Kennebec and Spunta) may affect yield and chemical composition of potato tubers, affecting the end use of the product. Flis [124] reported that the peculiarities of potato cultivar, plant root structure, and timing of nutrient uptake impact on the selection of a site-specific fertilization regime. Trehan et al. [125] showed that some cultivars exhibit strong symptoms of N, P and K deficiencies compared to others. Potato cultivars may sustain leaf development and nutrient uptake while maintaining maximum tuber growth rates to reach higher final tuber yields with contrasting nutrient requirements [126]. Differential

effects of cultivar and fertilizer on tuber yield have also been reported by Daoui et al. [127]. In a previous study, Coulibali et al. [128] found that genetic traits were not compelling to set apart clusters of cultivar based on N, P, K, Mg and Ca compositions of diagnostic leaves. The cultivar effect was thus excluded from the present study to keep models parsimonious. In our analysis, we focused on the gradients of N, P and K doses while keeping the other site-specific factors constant. Nevertheless, predictive features such as biotic factors (length of growing season, preceding crop, and seeding density), could also be predicted and optimized by the models with respect to tuber yield and quality.

## 4.2 Comparison of models

The performance of a predictive model is evaluated at testing or with unseen data set. The goodness of fit refers to how closely the model-predicted values match the true or observed values. Overfitting occurs where models perform well at training and badly at testing, while underfitting characterizes a model performing badly in both training and testing. Except for the Mitscherlich model, the model scores at testing showed discrepancies with training, reflecting problems of overfitting. The differences between $R^2$ values were highest for the marketable yield prediction models (Fig 3), reaching 0.40 with KNN. Based on those gaps, one could argue that our models did not generalize well from training to testing data. However, we used a robust approach by comparing different algorithms, tuning the hyperparameters and tuning the models using 5-fold cross-validation. The $R^2$ values at testing varied with respect to target variables but were practically similar between models. The models estimated the proportions of medium- and small-size tubers ([S | M] balance) more accurately than those of large-size tubers ([M, S | L] balance), probably because of the high number of zero weight values among large-size tubers (21%) compared to tubers of small (0.06%) and medium (0.4%) size, at the early stage of our analysis. Imputing zeros to deal with measures where the large size was completely absent [74] improved the prediction quality of this fraction. Except for the Mitscherlich model in predicting yield, the $R^2$ values at testing were greater than 0.50 and could be considered acceptable according to Moriasi et al. [92] for complex systems.

The Mitscherlich model returned a lower coefficient of determination in tuber yield prediction and was discarded for quality analysis (tuber size balances and SG). The KNN, RF, NN and GP algorithms more accurately approximated the unknown functions explaining tuber yield given the predictive features. However, it was difficult to select the best model since scores were practically similar. Cerrato and Blackmer [129] and several others [130–134] described similar ambiguities using classical statistical models.

Figs 4–7 indicated that the calibration and generalization procedures returned smooth response curves for the Mitscherlich, NN and GP models for all the target variables. Except for the low $R^2$ value of the former, the NN and GP models appeared more suitable for making inferences.

The prediction of optimum fertilizer doses and optimum or maximum outputs showed some disagreements for the case presented (Fig 8). There should be a single economic optimal dose or agronomic optimal dose at each site each year. Some models were more consistent than others in deriving optimal doses depending on the target variable. At extremely low predicted N, P or K doses, it could be challenging to manage the fertilization program at low economic risk for producers, who generally consider that the cost of over-fertilization is low compared to the cost of under-fertilization [37, 38]. The probabilistic prediction capability of Gaussian processes may help to determine credible dosage.

## 4.3 Probabilistic predictions

Sampling from a Gaussian process looks like rolling a die, returning a different function each time. Figs 9–12 showed only five possible functions for each target variable. By sampling the

process numerous times, we generated a distribution of economic or agronomic optimal fertilizer doses as those shown by the histograms of the figures. The distributions often show frequent optima at the edges to the NPK grid, *i.e.*, at dose of 0 or 250 kg ha$^{-1}$. This phenomenon emerges from sampling continuously increasing or decreasing GP samples, which are more frequent when the sample is close to patterns in data where the response to fertilizer is flat. A zero-fertilizer recommendation could be interpreted as a soil sufficiently fertile to supply the crop, or a soil poorly responsive due to other constraints [135] such as pests and diseases [20, 136] or weed damage [137]. Nevertheless, we covered a wide range of factors that may impact potato crop growth and yield without falling into mechanistic modeling. Fertilizer doses more than 250 kg ha$^{-1}$ may be excessive, since the maximum limits according to local standards are 175 kg ha$^{-1}$ for N, 87 kg ha$^{-1}$ for P and 199 kg ha$^{-1}$ for K [96].

To face predictions falling at the edges, the optimal fertilizer dosage could be selected within a range of conditional expectation as processed by Khiari et al. [43] when defining P optimal dose for acid coarse-textured soils. The $x^{th}$ conditional expectation dose is the optimal dose that produces optimal yield *x%* of the time. For example, the 60$^{th}$ percentile would be the sampled optimal dose that produces optimal yield 60% of the time for a given site. Khiari et al. [43] assessed the 50$^{th}$ and 80$^{th}$ percentiles. The mean (50%), the median or any other percentile dose could be computed to support decision-making. For example, the mean GP and the probability distribution processes returned the upper bound of the simulation dosage (*i.e.*, 250 kg N ha$^{-1}$) as the economic optimal dose for the N trial with the marketable yield prediction model (Fig 9). The conditional expectation percentiles showed that a lower dose (*i.e.*, 223 kg N ha$^{-1}$) could be recommended, producing optimal yield 55% of the time. At the 60$^{th}$ percentile or more, the full dose *i.e.*, 250 kg N ha$^{-1}$ must be applied.

## 5. Conclusion

This study assessed machine learning techniques as an alternative for potato fertilizer recommendations at local scale usually handled by statistical models or meta-analysis at regional scale. A large collection of field trial data provided information to fit machine learning models with specific traits of cultivars, soil properties, weather indexes, and N, P and K fertilizers dosage used as predictive features. Five models, Mitscherlich, KNN, RF, NN and GP, were evaluated against optimal economic N, P and K doses derived from yield, or against optimal agronomic N, P and K doses derived from tuber size and SG. The models trained using machine learning algorithms outperformed the Mitscherlich tri-variate response predictive model. The marketable yield prediction coefficient ($R^2$) varied between 0.49 and 0.59, while the Mitscherlich model returned 0.37. The large-size tuber balance was predicted with a coefficient varying between 0.55 and 0.64. The $R^2$ varied between 0.60 and 0.69 in predicting medium-size tuber balance, and between 0.58 and 0.67 for SG. The N, P and K optimal doses could be recommended with respect to marketable yield, tuber size or SG using the NN and GP models, which appeared to be the most suitable for making inferences. Response surfaces were obtained by conditioning the models using N-P-K doses generated from uniform distributions under constant weather conditions, soil properties and land management factors. The GP model stood up by its probabilistic framework in risk estimation for potato fertilizer recommendation in Quebec conditions.

As large amounts of data are being assembled into observational data sets, machine learning models may surrogate statistical models in making fertilizer recommendations in the context of precision agriculture. To assess model performance under real-world situations, it was an effective strategy to combine historical weather data since accurate future weather data covering the growing season are unavailable. We also focused on using easily-available features

collected from routine analyses as predictors instead of mechanistic processes models. Any biotic factor other than fertilizer, *e.g.*, length of growing season or planting density, could be optimized with our model. Improvement will require more data from many more diverse environments and management scenarios. With more experiment data, the training and testing division could be performed at trial level to improve the model predictive ability. Moreover, since the data for this analysis were collected from small research plots, validation at production-scale fields is needed for decision making.

## Supporting information

**S1 Table. Description of the marketable yield modeling data set.**
(DOCX)

**S2 Table. Description of the data sets used for modeling per trial type.**
(DOCX)

**S3 Table. Classification of preceding crops [7].**
(DOCX)

**S4 Table. Centroids of soil textural classes derived from the Quebec soils data set [57].**
(DOCX)

**S5 Table. Quebec potato data set used for modeling.** 'Potato_df.csv' file available in 'data' repository at https://git.io/JvYxd.
(CSV)

## Author Contributions

**Conceptualization:** Zonlehoua Coulibali, Serge-Étienne Parent.

**Data curation:** Zonlehoua Coulibali, Serge-Étienne Parent.

**Formal analysis:** Zonlehoua Coulibali, Serge-Étienne Parent.

**Investigation:** Zonlehoua Coulibali, Athyna Nancy Cambouris, Serge-Étienne Parent.

**Methodology:** Zonlehoua Coulibali, Serge-Étienne Parent.

**Project administration:** Serge-Étienne Parent.

**Resources:** Serge-Étienne Parent.

**Software:** Zonlehoua Coulibali, Serge-Étienne Parent.

**Supervision:** Serge-Étienne Parent.

**Validation:** Zonlehoua Coulibali, Serge-Étienne Parent.

**Visualization:** Zonlehoua Coulibali, Serge-Étienne Parent.

**Writing – original draft:** Zonlehoua Coulibali, Athyna Nancy Cambouris, Serge-Étienne Parent.

**Writing – review & editing:** Zonlehoua Coulibali, Athyna Nancy Cambouris, Serge-Étienne Parent.

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
