## [Decision Letter · Decision Letter 0]

6 Apr 2020

PONE-D-20-06893

Site-specific machine learning predictive fertilization models for potato crops in Eastern Canada.

PLOS ONE

Dear Prof. Parent,

Thank you for submitting your manuscript to PLOS ONE. After careful consideration, we feel that it has merit but does not fully meet PLOS ONE’s publication criteria as it currently stands. Therefore, we invite you to submit a revised version of the manuscript that addresses the points raised during the review process.

We would appreciate receiving your revised manuscript by May 21 2020 11:59PM. To enhance the reproducibility of your results, we recommend that if applicable you deposit your laboratory protocols in protocols.io, where a protocol can be assigned its own identifier (DOI) such that it can be cited independently in the future. For instructions see: http://journals.plos.org/plosone/s/submission-guidelines#loc-laboratory-protocols

We look forward to receiving your revised manuscript.

Kind regards,

Vassilis G. Aschonitis

Academic Editor

PLOS ONE

Journal Requirements:

Reviewers' comments:

Reviewer's Responses to Questions

**Comments to the Author**

1. Is the manuscript technically sound, and do the data support the conclusions?

Reviewer #1: Partly

2. Has the statistical analysis been performed appropriately and rigorously? 

Reviewer #1: No

3. Have the authors made all data underlying the findings in their manuscript fully available?

Reviewer #1: Yes

4. Is the manuscript presented in an intelligible fashion and written in standard English?

Reviewer #1: Yes

5. Review Comments to the Author

Reviewer #1: “The authors made an interesting work in assessing site-specific learning predictive fertilization models for potato crops in Eastern Canada. They used multiple data and variables trying to assess different prediction methods regarding tuber yield and potato quality (also N,P,K requirements for potato yield and quality). However, the data are not presented clearly and in detail (no descriptive statistics) so we don't know if they are the appropriate for the current models. The authors tried to combine data from different time periods (from the last 40 years) something that can also be problematic and needs more clarification. Finally, they tried to assess multiple things at the same time. This confuses the reader and perplexes the understanding of the goals and the scope of the paper. Therefore, we believe that the manuscript needs major revisions before it could be accepted for publication”

The authors should consider the following:

1. Please provide detailed descriptive statistics for all the data that you used. Also provide measure of units, small description, methods of analysis and time (date) of measurements for your data. The current tables in the manuscript are more confusing than helpful.

2. The 273 field experiments that you use extend in a very long time period (from 1979 to 2017). Are you sure that the data from these experiments are consistent (e.g. same methods of collecting and measuring the parameters) so to be used in data analysis.

3. How are the field experiments spread throughout the years? Provide at least some kind of classification with percentages (1979-1989 20%, 1990-1999 30% etc.).

4. It is unclear how you combined the data from tables 1,2,3,4,5 and 6. For example, you may have data from a field experiment that happened 40 years ago and weather historical data from 5 years ago (line 222). Can you use them together? Also, please provide the time (date) of the collection and analysis of the soil profiles (and soil samples)? Are they consistent with the timeframe of the field experiments and weather data?

5. You have the geographical coordinates from each experiment. It would be helpful if you provide a map with the locations of the experiments?

6. You say that you added data from another 17 trials (line 124), in the 273 field experiments. How many are the field experiments overall and exactly how many are the samples for each parameter used in the models?

7. There seem to be a conflict between the objectives of the study. In lines 21-24 you say that you want to predict tuber yield and quality, whereas in lines 95-97 you state that you want to compare the performance of ML in predicting N,P,K requirements for potato yield and quality. Please make totally clear what you try to achieve. Also, try to keep it simple and to the point. Too much goals, confuse and distract.

8. Please change "neuronal networks" to "neural networks" (e.g. In line 27 and line 284)

9. What type of random cross validation did you use? (e.g. leave one out, k-fold..how many folds?)

10. Check the "Error! Reference source not found" error

11. The tuning parameters (lines 361-374) are in the wrong place ( 4.1 Feature importance). Please place them in a separate section.

12. Images in manuscript are not clear. They are have a very low quality.

13. Too long image captions. E.g in lines 399-401 there is no need to explain what MAE, RMSE etc are.

14. In line 434 too many "and". Please rephrase.

15. Please mention the prediction method/s that the feature importance (line 349) belongs?

16. In line 560, the discrepancies in scores between testing and training data shows problems of overfitting or underfitting?

17. The text that you use for the hyperparameters/parameters is confusing (lines 361-374). Please provide also a table/s with the hyperparameters/parameters of the models, a small description and the values that you used.

6. PLOS authors have the option to publish the peer review history of their article (what does this mean?). If published, this will include your full peer review and any attached files.

Reviewer #1: No

---

## [Author Response · Author response to Decision Letter 0]

21 May 2020

Dear reviewer(s),

We revised the manuscript (PONE-D-20-06893) entitled “Site-specific machine learning predictive fertilization models for potato crops in Eastern Canada” according to the reviewers’ comments. We thank the referees for their careful and constructive reviews. The responses are detailed according to the progression of the reviewer’s notes in the file named Response to Reviewers.docx.

Thank you again for your consideration of our revised manuscript.

Best regards,

Serge-Étienne Parent, Eng, PhD

Assistant Professor, Laval University 

Faculty of Agriculture and Food Sciences

---

## [Decision Letter · Decision Letter 1]

16 Jun 2020

PONE-D-20-06893R1

Site-specific machine learning predictive fertilization models for potato crops in Eastern Canada

PLOS ONE

Dear Dr. Parent,

Thank you for submitting your manuscript to PLOS ONE. After careful consideration, we feel that it has merit but does not fully meet PLOS ONE’s publication criteria as it currently stands. Therefore, we invite you to submit a revised version of the manuscript that addresses the points raised during the review process.

We look forward to receiving your revised manuscript.

Kind regards,

Vassilis G. Aschonitis

Academic Editor

PLOS ONE

Reviewers' comments:

Reviewer's Responses to Questions

**Comments to the Author**

1. If the authors have adequately addressed your comments raised in a previous round of review and you feel that this manuscript is now acceptable for publication, you may indicate that here to bypass the “Comments to the Author” section, enter your conflict of interest statement in the “Confidential to Editor” section, and submit your "Accept" recommendation.

Reviewer #2: All comments have been addressed

2. Is the manuscript technically sound, and do the data support the conclusions?

Reviewer #2: Yes

3. Has the statistical analysis been performed appropriately and rigorously? 

Reviewer #2: Yes

4. Have the authors made all data underlying the findings in their manuscript fully available?

Reviewer #2: Yes

5. Is the manuscript presented in an intelligible fashion and written in standard English?

Reviewer #2: Yes

6. Review Comments to the Author

Reviewer #2: The author has improved this work well under the comments of the first external review expert. The article is very informative and becomes more meaningful. The data is also supplemented. The article uses multiple statistical models and also adds a description. It was explained. However, we believe that there are several points in this article, which we hope can be improved:

Although the 40-year change mentions the improvement of the measurement method, etc., it is possible to discuss more about whether the variety has the property of preferring a certain fertilizer or the relationship between the varieties.

7. PLOS authors have the option to publish the peer review history of their article (what does this mean?). If published, this will include your full peer review and any attached files.

Reviewer #2: No

---

## [Author Response · Author response to Decision Letter 1]

8 Jul 2020

Reviewer #2

The author has improved this work well under the comments of the first external review expert. The article is very informative and becomes more meaningful. The data is also supplemented. The article uses multiple statistical models and also adds a description. It was explained. However, we believe that there are several points in this article, which we hope can be improved:

Although the 40-year change mentions the improvement of the measurement method, etc., it is possible to discuss more about whether the variety has the property of preferring a certain fertilizer or the relationship between the varieties.

Answer

We have amended the discussion about the selection of predictive features in the last paragraph of the 5.1 section (Selection of features):

The relative importance of a variable in a model is related to its effect on the output through its gradient in the data set. Hence the predominance of N doses, and P and K doses to some extent, could have been caused by the origin of the data set, which is a collection of fertilizer trials where large gradients of doses are found by design. This study did not address fertilizer source and timing of application. While Marouani et al. [122] found equivalency of ammonium nitrate (33.5% N), urea (46% N), NP fertilizer (33% N – 14% P2O5) and NPK fertilizer (27%N – 5% P2O5 – 5% K2O), Petropoulos et al. [123] found that the form of the fertilizer (ammonium sulfate, ammonium sulfate + zeolite, manure, slow release N fertilizer with urease inhibitor) and the cultivar (Kennebec and Spunta) may affect yield and chemical composition of potato tubers, affecting the end use of the product. Flis [124] reported that the peculiarities of potato cultivar, plant root structure, and timing of nutrient uptake impact on the selection of a site-specific fertilization regime. Trehan et al. [125] showed that some cultivars exhibit strong symptoms of N, P and K deficiencies compared to others. Potato cultivars may sustain leaf development and nutrient uptake while maintaining maximum tuber growth rates to reach higher final tuber yields with contrasting nutrient requirements [126]. Differential effects of cultivar and fertilizer on tuber yield have also been reported by Daoui et al. [127]. In a previous study, Coulibali et al. [128] found that genetic traits were not compelling to set apart clusters of cultivar based on N, P, K, Mg and Ca compositions of diagnostic leaves. The cultivar effect was thus excluded from the present study to keep models parsimonious. In our analysis, we focused on the gradients of N, P and K doses while keeping the other site-specific factors constant. Nevertheless, predictive feature such as biotic factors (length of growing season, preceding crop and seeding density), could also be predicted and optimized by the models with respect to tuber yield and quality.

After proofreading, this suggestion helped us significantly improve the section and the manuscript.

---

## [Decision Letter · Decision Letter 2]

21 Jul 2020

Site-specific machine learning predictive fertilization models for potato crops in Eastern Canada

PONE-D-20-06893R2

Dear Dr. Parent,

We’re pleased to inform you that your manuscript has been judged scientifically suitable for publication and will be formally accepted for publication once it meets all outstanding technical requirements.

Kind regards,

Vassilis G. Aschonitis

Academic Editor

PLOS ONE

Additional Editor Comments (optional):

Reviewers' comments:

Reviewer's Responses to Questions

**Comments to the Author**

1. If the authors have adequately addressed your comments raised in a previous round of review and you feel that this manuscript is now acceptable for publication, you may indicate that here to bypass the “Comments to the Author” section, enter your conflict of interest statement in the “Confidential to Editor” section, and submit your "Accept" recommendation.

Reviewer #2: All comments have been addressed

2. Is the manuscript technically sound, and do the data support the conclusions?

Reviewer #2: Yes

3. Has the statistical analysis been performed appropriately and rigorously? 

Reviewer #2: Yes

4. Have the authors made all data underlying the findings in their manuscript fully available?

Reviewer #2: Yes

5. Is the manuscript presented in an intelligible fashion and written in standard English?

Reviewer #2: Yes

6. Review Comments to the Author

Reviewer #2: (No Response)

7. PLOS authors have the option to publish the peer review history of their article (what does this mean?). If published, this will include your full peer review and any attached files.

Reviewer #2: No

---

## [Editor Report · Acceptance letter]

24 Jul 2020

PONE-D-20-06893R2 

Site-specific machine learning predictive fertilization models for potato crops in Eastern Canada 

Dear Dr. Parent:

I'm pleased to inform you that your manuscript has been deemed suitable for publication in PLOS ONE. Congratulations! Your manuscript is now with our production department. 

Kind regards, 

on behalf of

Dr. Vassilis G. Aschonitis 

Academic Editor

PLOS ONE